# Assessment of the Interaction of the Combined Piled Raft Foundation Elements Based on Long-Term Measurements

**DOI:** 10.3390/s25113460

**Published:** 2025-05-30

**Authors:** Grzegorz Marek Kacprzak, Semachew Molla Kassa

**Affiliations:** 1Faculty of Civil Engineering, Warsaw University of Technology, 00-637 Warsaw, Poland; 2College of Engineering and Technology, Wachemo University, Hossana P.O. Box 667, Ethiopia; smakmolla23@gmail.com

**Keywords:** combined pile raft foundation, long-term in situ measurements, FEM analysis, sensors

## Abstract

Understanding the complex phenomena of interactions between the elements of a combined piled raft foundation (CPRF) is essential for the proper design of such foundations. To evaluate the effects of mutual influence among the CPRF’s elements, a series of long-term measurements of selected physical quantities related to the performance of the foundation were conducted on a building with a frame structure, stiffening walls, and monolithic technology, consisting of seven aboveground stories and one underground story. The analysis distinguishes the real deformations resulting from temperature changes and from stress strains resulting from load changes. The two types of deformations were subjected to further interpretation of only changes in the stress and strain over time. Changes in stress values in the subsoil, as well as strain measurements in the vertical direction of concrete columns, were recorded to assess the load distribution between the CPRF’s components. The numerical analysis results obtained for a fragment of the monitored foundation were compared with actual measurement results to verify the numerical model of interaction between the structure and the soil. Field monitoring and FEA methods were used to compare the long-term deformation analysis, and they helped to minimize the monitoring time. This comparison also served to supplement and simultaneously expand the dataset of test results on a real-world scale.

## 1. Introduction

Gotman N.Z. and Gotman A.L. [1] analyzed the interaction between piles, soil, and slabs in a combined piled raft foundation in cohesive soils, focusing on changes in axial forces within the piles. Measurements of axial force changes were taken using strain-gauge-equipped force meters placed at the base and head of 9.0 m long piles, as well as at two cross-sections located at depths of 4 m and 7 m below the pile head. The study considered three configurations: piles working individually, piles in a group, and piles in a combined piled raft foundation. Their findings showed that in a combined piled raft foundation, pile–pile interaction causes the mobilization of friction on the pile side only in the lower portion—between 25% and 33% of the total pile length measured from the bottom. This contrasts with piles working alone, where friction is mobilized along the entire pile length from top to bottom.

Oliver Reuil and Mark F. Randolph [2] studied the behavior of the combined piled raft foundation of the Westend Duo building in Frankfurt, which was founded on Frankfurt clay. The foundation slab covered an area of 4100 m^2^, and monitoring included measurements of axial forces in three piles, stress on the ground beneath the slab using five pressure sensors, groundwater pore pressure using five sensors, and settlement at 23 points using geodetic surveying. The authors reported that the maximum settlement of the slab, 4.7 cm, did not exceed the estimated value of 7.2 cm. Preliminary analyses had indicated that the piles would absorb approximately 40% of the total load (αFPP = 40%).

Kharichkin [3] focused on measuring vertical forces in piles to evaluate their role as “reducers” of settlement in a raft footing under a residential building in Moscow. The building consisted of two sections, one 70 m tall (22 stories) and the other 90 m tall (30 stories), with a total foundation area of 3000 m^2^. Each section was supported by driven prefabricated piles spaced at 0.9 × 0.9 m and 1.05 × 1.05 m, respectively, with lengths of approximately 14 m and cross-sectional dimensions of 0.3 × 0.3 m. Six piles were equipped with SVD-100 force sensors welded to steel plates at the pile heads to measure forces. Results indicated that corner piles experienced forces of 345 kN, which were 1.9 times greater than piles along the edge of the slab and 2.3 times greater than piles located within the slab’s plan. Settlements were measured as 18.6 mm for the lower building and 21.0 mm for the taller building [4].

Ryuuichi Sonoda [5] analyzed a building in Port Island, Kobe City, founded on a pile–slab foundation. The seven-story steel-frame building, with dimensions of 35.25 × 55.80 m^2^ in plan, was supported by a 0.4 m thick slab and 2.4 m high foundation beams resting on 93 prestressed high-strength concrete piles (PHC) with a diameter of 0.65 m and a length of 12 m. One pile was equipped with deformation sensors at depths of 4.4 m, 8.4 m, and 11.4 m. The monitoring system also included 12 benchmarks for geodetic settlement measurements and a point for measuring pore water pressure in the ground. The forces measured in the piles after construction closely matched numerical predictions. The safety of different structures is monitored using a field monitoring system and another monitoring system [6].

In Frankfurt am Main, Katzenbach [7] carried out automatic measurements of piles and subsoil for high-rise buildings founded on combined piled raft foundations in Frankfurt clay during the 1990s and subsequent years. These measurements verified the numerically estimated load distribution between piles and subsoil under foundation slabs. One notable example is the Messeturm building, which is 256 m tall and founded on a square plan of 58.8 m × 58.8 m. Its foundation consists of a 6.0 m thick slab in the central area, tapering to 3.0 m at the edges, supported by 64 bored piles with diameters of 1.3 m and lengths ranging from 26.9 m to 34.5 m, spaced 3.5 to 6.0 diameters apart. The 256 m high Messeturm building, on a square plan of 58.8 m × 58.8 m, founded on a 6.0 m thick slab in the central prt, decreasing to 3.0 m at the edges, and on 64 piles of 1.3 m in diameter and lengths ranging from 26.9 m to 34.5 m, spaced 3.5 to 6.0 diameters, was described in detail by Sommer et al. [8], Sommer [9], El-Mossallamy and Franke [10], and Tamaro [11]. Monitoring included geodetic measurements of settlements and tilts, ground settlement at various depths using extensometers, ground stresses under the slab using pressure sensors, and axial force distribution along piles using deformation measurements at six points. These observations revealed that 55% of the load was carried by the piles, while 45% was borne by the ground under the foundation slab. Additionally, friction mobilized by the piles varied with depth, ranging from 25 to 70 kPa at depths up to 15 m below the slab base and increasing to 110 kPa for inner piles and 160 kPa for outer piles at greater depths. Katzenbach found that piles in combined piled raft foundations mobilized significantly greater friction compared to piles working individually.

Similar monitoring was conducted for the Messe Torhaus building in Frankfurt am Main, which is approximately 130 m tall (30 floors). The building was supported by two independent slabs, each resting on 42 bored piles with a diameter of 0.9 m and a length of 20 m, spaced 3–3.5 diameters apart. Measurements showed that 80% of the load was carried by the piles, while 20% was borne by the subsoil under the slab. The ground’s load contribution originated primarily from the self-weight of the foundation slab [12,13].

Katzenbach also analyzed the DG-Bank building (Westendstrasse 1 in Frankfurt am Main), which is 208 m tall with 53 floors. The foundation consisted of a 3.0–4.5 m thick slab covering an area of 2940 m^2^, supported by 40 bored piles with diameters of 1.3 m and lengths of 30 m. Measurements indicated that half of the load was carried by the ground under the foundation slab. Further studies by Katzenbach examined load distribution between piles and subsoil for other buildings founded on Frankfurt clay, including the Taunustor-Japan Centre (40% piles), Forum-Kastor and Pollux (35–40% piles), Congress Centre Messe Frankfurt (40% piles), Main Tower (85% piles), Eurotheum (30% piles), and Commerzbank (96% piles).

This work’s original contribution is to assess the long-term measurements of CPRF deformations using fieldwork measurements (sensors) and finite element methods (ZSoil version 2011) software.

The main objective of this study is to assess the interaction of elements constituting the combined piled raft foundation of an eight-story monolithic reinforced concrete building. A measurement system was designed and implemented for the first time in Poland. The system includes a method for measuring stress changes in the ground under the foundation slab and deformation measurements of foundation columns to estimate theoretical force changes at three depths.

## 2. Materials and Methods

### 2.1. Soil and Water Conditions

The analyzed building is located within a deep Carpathian foreland filled with Miocene sediments. In the ceiling of Miocene clays, which were not drilled directly under the building, an erosion incision was eroded and filled with silty periglacial formations several to a dozen or so meters thick. These formations were formed mainly in silty clays, silt, and fine and silty sands covered from the ground surface with fluvioglacial sands. Within the sandy-silt intercalations in Miocene formations, at a depth of 10 to 25 m below ground level, a Tertiary aquifer was drilled and taken in the area by drilled wells. Hydrated sand lenses were found with groundwater under low hydraulic tension within the periglacial deposits. Groundwater was observed in the near-surface layer of fluvioglacial sands during drilling.

Based on the research results, two essential geotechnical layers can be distinguished in the subsoil under the building, as shown in Figure 1. The first one is a layer of Quaternary Pleistocene silty periglacial formations, developed in the form of silty clays, silt, sandy silt, and less frequently clays, silty clays, sporadically sandy clays, and clayey sands, in a plastic and soft-plastic state with a liquidity index of IL = 0.4–0.55. Below are silty and fine sands, constituting the interbedding of Miocene clays in a compacted state, with a density index of approximately ID = 0.7–0.8. The red dot indicates the vertical profile of the soil corresponding to the boherholes.

The ground conditions in the vicinity of columns P1–P6 (see Figure 1) were detailed by adding the results from the closest CPT—Figure 2 for P1, from the dilatometric test (Figure 3 for column P3) and Figure 4 (for P1 to P6). The resistance of the CPT recorded at the level of 2.0–5.0 MPa confirms the state of silty soils that was macroscopically recognized and presented in the geological engineering cross-section (Figure 1) (by PN-B-04452: *IL* = 0.21 for *qc* = 5.0 MPa, *IL* = 0.51 for *qc* = 2.0 MPa).

### 2.2. Building Structure

The analyzed building has the shape of a cuboid with a rectangular base with dimensions of 26.90 m × 84.20 m.

The building has a height of 32.6 m from the foundation level, consisting of seven aboveground floors and one underground floor located in the “deeper” section. The facility was designed using monolithic construction technology as a skeleton structure with stiffening walls. The building is founded 4.50 m below ground level on a foundation slab that is 0.80 m thick, with localized thickening to 1.0–1.2 m under the pillars. Table 1 provides the start dates for the individual concreting stages as well as the indicative layout of the floors in the analyzed building. To limit vertical displacements of the structure, localized deep-ground reinforcement was implemented in the area where pillars and walls meet the foundation slab. This reinforcement was carried out using CMC concrete displacement columns (Les Colonnes à Module Contrôlé), which have a diameter of 0.40 m. The columns were embedded into the load-bearing soil, consisting of silty sands, to a depth of at least 1.0 m. The total length of the CMC columns ranged from 8.4 m to 19.3 m. For further analysis, the foundation system consisting of the CMC columns and the soil beneath the slab was modeled and analyzed as a combined piled raft foundation (CRPF).

The monitored building is part of the newly constructed hospital in Kraków-Prokocim, which consists of a total of twenty facilities, including nine hospital buildings with a combined total area of 150,000 m^2^ and a volume of 500,000 m^3^. The construction was completed within 4.5 years. The analyzed building was monitored for almost 800 days (which corresponds to approximately 80% of the total planned load (excluding variable load from crowds). The shell stage of the building was completed after approximately 200 days, reaching 50% of the total planned load (see Table 1).

### 2.3. Measurements of Deformation of CMC Concrete Displacement Columns

The measurements included six CMC concrete displacement columns, where longitudinal deformations were measured at three depths. Each column, located beneath the recesses of the foundation slab, has a diameter of 0.40 m and extends several meters in length. The columns were embedded through layers of weak-bearing silt and dust and rested in a layer of sand, penetrating the sand layer to a depth of at least 1.0 m. The locations of the measured columns and the adopted nomenclature are illustrated in Figure 5. In the subsequent sections of the paper, the axial force variation was analyzed for all columns (from P1 to P6). Additionally, a stress distribution analysis was performed for the slab–subgrade–column system near column P1 [16], focusing on the distribution of load transfer between the structure, the soil beneath the foundation slab, and the concrete column(s).

#### 2.3.1. Measuring Sensors

The system was constructed using Geokon model 4800 wire pressure sensors, which have a measurement range of 170 kPa. These sensors consist of two stainless steel sheets welded together around their perimeter, with a narrow gap between them filled with hydraulic fluid [17,18]. External loads applied to the sensor surface result in changes in the volume of the chamber, causing the fluid to move into or out of the pressure transducer. This movement alters the length of the string connected to the steel membrane that is in contact with the fluid, thereby changing the string’s natural frequency [19,20,21]. Assuming the hydraulic fluid is incompressible, the difference in the frequency of the string vibrations can be translated into changes in pressure [22,23]. Integration with a temperature sensor enables thermal compensation for the sensor readings, ensuring accuracy across varying temperatures. Using individual calibration cards, calculations are performed to determine changes in vertical stresses in the ground beneath the foundation slab, expressed directly in kPa [24,25]. Figure 6 illustrates the construction details of the measuring device, while its basic technical data are provided in Table 2.

#### 2.3.2. Installation Method

The pressure transducers were installed in the ground beneath the leveling concrete. Prior to installation, the mounting locations were carefully prepared: the ground was excavated to a depth of approximately 0.10 m, and the shape of the recess was adjusted to match the dimensions of the sensor, as shown in Figure 7. Each transducer was placed inside a polypropylene woven bag filled with sand. This procedure was designed to eliminate point interactions and ensure uniform pressure transfer from the foundation slab to the sensor.

### 2.4. Measurements of Stresses in the Ground Under the Foundation Slab

To estimate changes in stress values in the ground at the contact point with the foundation slab caused by the varying self-weight of the building during construction, three wire pressure transducers (measured in kPa) were installed in the equalization chamber [26]. One of the sensors was placed in the depression near the measured concrete column (S8), while the other two were positioned as follows: one in the middle of the distance between the edges of adjacent depressions (S7), and the other in the middle of the distance between the edge of the depression and the nearest concrete column (S9), as shown in Figure 8.

#### 2.4.1. Method of Measuring Deformations

In each of the measured concrete columns, string sensors were installed at three different depths to measure the vertical deformation of the shafts (expressed in microstrain [µε]). These sensors enabled the observation of changes in force distribution along the column due to variations (or consistency) in the applied load [27].

Steel profiles were embedded within the analyzed columns, and Geokon model 4100 (Geokon, Inc., Lebanon, NH, USA), wire strain gauges were heated and attached to the profiles at three designated locations. Figure 9 provides a visual representation of the measuring device, while its basic technical data are summarized in Table 3.

The sensors are enclosed within steel protective housings, as shown in Figure 10, to prevent damage during installation. These housings are specifically designed to minimize resistance from the concrete mix during the pressing or vibrating of the steel profiles, as illustrated in Figure 10, Figure 11, Figure 12 and Figure 13.

Columns P1 and P2 were equipped with steel pipes measuring approximately 50 mm in diameter, with a wall thickness of 2.9 mm and a length of 15.2 m. The locations of the sensors along the length of the profile are shown in Figure 14. After subjecting these columns to test loads, the upper part of the profile, approximately 1.8 m in length, was cut off to align with the upper surface of the base concrete [28].

Columns P3, P4, P5, and P6, on the other hand, were equipped with IPE120 steel I-beams (TD Slavsant, Kyiv, Ukraine), each 12.0 m long. Figure 14 also illustrates the sensor placements along the length of these profiles.

#### 2.4.2. Assessment of Compressive Forces in CMC Columns

The determination of compressive forces in columns based on deformation measurements was carried out based on physical laws relating stress to deformation. For the purposes of this study, it was assumed that the dependency is linear, where Ncol is the normal force of the column, Acs is cross-sectional area of the column, Ecm is the elastic modulus of the sandstone aggregate, ε is the change in deformation measured by the sensor, and σ is the stress.(1)$$\sigma =E_{cm}\varepsilon$$

The value of the modulus of elasticity, *E*, was adopted based on the PN-EN 1992-1-1:2008 standard Eurocode 2—Design of concrete structures—Part 1-1: General rules and rules for buildings [29] for concrete class C25/30, reducing it by 30% due to the use of sandstone aggregate: Ecm = 21.7 GPa.

It was assumed that there is full adhesion between the steel profile and the concrete, ensuring the compatibility of deformations. The transformed cross-sectional areas *Acs* were determined for columns with an IPE120 profile and a tubular profile, taking into account the ratio of the modulus of elasticity of steel Est = 200 GPa and concrete Ecm = 21.7 GPa.

The value of the force Ncol in the column at various depths was determined using the formula(2)$$N_{col}={A_{cs}E}_{cm}\varepsilon$$
where ε is the change in deformation measured by the sensor, taking into account the correction related to different thermal expansion coefficients of steel and concrete, which corresponds to the stresses in the structural element, excluding free thermal deformations (as further explained in the text).

It should be noted that the values of the forces were determined under the assumption of constant stiffness in compression of the column along its length (AcsEcm = const). This means that the assumption was made that along the length of the column, there are no disturbances causing a change in the column’s diameter (Acs = const) or a change in the elastic properties of the column concrete (Ecm = const).

Based on the above detailed introduction and the field tests incorporated in this section, we have provided the complete workflow in this paper, as shown in Figure 15.

## 3. Results and Discussion

### 3.1. Take-Off Measurements of Column Deformation

Take-off measurements (“zero measurements”) for the deformation sensors installed in the columns were taken immediately after their completion, i.e., on 12 October 2015. The readings of the sensors in columns P1 and P2 were also read during the test load. The detailed procedure is shown in Section 3.2.

Automatic measurements were started on 20 May 2016. In Figure 16 and Figure 17, the recorded strain values from this period are presented in relation to the assumed take-off measurements from 12 October 2015. The dotted lines in Figure 16 and Figure 17 were introduced only to increase the figures’ readability because the intermediate periods’ strain functions are unknown.

Temperature was measured at every step and was taken into account when determining e(s)—stress and strain values–as part of the true stresses (line 462). It cannot be clearly determined whether the first measurement, taken on 12 October 2015, reflects the temperature increase due to the heat of hydration or the cooling phase of the concrete. Comparing the measured temperatures of 20.3 °C, 18.4 °C, and 20.7 °C recorded on 12 October 2015, in sensors P1/1, P1/2, and P1/3, respectively, with the values recorded a month later—11.2 °C, 11.4 °C, and 11.7 °C—it can be observed that the temperature of the concrete decreased along the entire length of the column. On 20 May 2016, the temperature stabilized across all sensors at 9.9 °C, 9.0 °C, and 11.6 °C and remained exactly the same one year later. In column P2, a temperature decrease over time was observed in all sensors. The recorded values for sensors P2/1, P2/2, and P2/3 were, respectively, 15.7 °C, 16.4 °C, and 15.5 °C on 12 October 2015; 10.2 °C, 10.2 °C, and 10.1 °C on 12 November 2015; and 9.2 °C, 8.9 °C, and 10.9 °C on 20 May 2016. Finally, it should be concluded that the concrete temperature measurements indicate a cooling trend over time.

Based on the graphs of deformations measured in columns P1 and P2 at three depths (Figure 16 and Figure 17), from their installation on 12 October 2015 until the activation of automatic measurements on 20 May 2016, three distinct phases can be identified. The first phase corresponds to changes in deformation due to the maturation of the concrete mix. The second phase is associated with deformation changes resulting from the application and removal of the test load. The third phase covers the waiting period for the initiation of automatic measurements during the building’s construction, reflecting an increase in load due to the building’s own weight.

In the first phase, column shortening was observed in all sensors except P1_3, most likely caused by temperature changes following the release of heat during cement hydration, along with concrete shrinkage. Shrinkage occurs due to structural changes in the cement paste caused by physicochemical processes of water loss during concrete setting and hardening. It is assumed that shrinkage was facilitated by the soil’s moisture content of 25%. In contrast, sensor P1_3, located near the column head, exhibited an opposite trend: swelling, likely caused by the presence of water in the upper soil layers during the setting of the concrete mix. Column shortening may also have resulted from the load imposed by the concrete mix itself. However, since column deformations were measured at only two points in time (at the start of the concrete setting process and at the onset of the test load), it is not possible to fully or accurately interpret the processes occurring in the column during the initial 30 days after its installation in the ground.

The strain graphs for both columns during the load tests are presented in Figure 18a and Figure 19a. In this analysis, the zero strain reading corresponds to the measurement taken prior to the application of the test load. The recorded strain values reflect only the changes resulting from the variation in the load applied to the column head.

During the second phase, corresponding to the test load, column shaft shortening was observed (Figure 19), accompanied by a residual force remaining in the column shaft after unloading. Based on deformation measurements, it can be concluded that the residual force in column P1 is greater than that in column P2. Comparing these observations with the axial force changes along the column during the test load (Figure 20), these conclusions appear well founded. The performance of the side surface of column P1 is better compared to column P2, which ultimately results in the mobilization of greater force during the test load and, consequently, a higher residual force in column P1.

In column P2 (Figure 20b), during the first stage of loading, the force estimated from the measurement with sensor P2_3 was only slightly higher than the force estimated using sensor P2_2. After the fifth stage and unloading of the column, the force at point P2_2 was higher than that at point P2_3 during the subsequent loading step. Although a measurement error cannot be ruled out, the author believes that this axial force distribution could indicate narrowing in the vicinity of sensor P2_2. It may also suggest the absence of friction mobilization on the column side up to the depth at which the P2_2 sensor was located.

In the third and final phase, a phenomenon similar to that observed in the first phase is evident. Measurements from all sensors, except P1_3, indicate column shortening, while sensor P1_3 shows column lengthening. As in the first phase, this can be attributed to the presence of water in the substrate, which causes soil swelling—a phenomenon observed exclusively in the upper part of the column.

### 3.2. Load Test of Columns P1 and P2

Test Procedure

The tests were conducted from 4 November to 17 November 2015.

The test loading was performed in accordance with PN-83/B-02482 regarding the bearing capacity of piles and pile foundations [30].

To conduct the tests for six columns, a structure composed of a main beam (2×IPE550) with a length of 8 m and two secondary beams (2×HEB300) with a length of 4 m was used. The connection between the secondary beams and the anchoring piles was made by welding, as specified in the design documentation (PW).

The load application was performed using a setup consisting of (a) one hydraulic cylinder (serial number 1) with a nominal force of 2000 kN, and an electric-powered hydraulic pump with a pressure gauge; (b) the monitoring of the column displacement using three dial gauges with a range of 50 mm and a readout accuracy of 0.01 mm; (c) a precision leveling method for controlling the displacement of the anchoring columns.

The test loads were performed in two stages:

Stage 1:

The load was incrementally increased until reaching a force approximately equal to the calculated design load (Q).

After reaching the force close to the calculated design load (Q), the column was unloaded to zero, and the permanent deformation was stabilized for 10 min.

Stage 2:

Following the reinforcement designer’s suggestion, the column was loaded to a force equal to 200% of the calculated design load. The column was then unloaded, and after 10 min, the permanent deformation was recorded. The image in the Table 4 and Table 5 are used to measure the displacement. The data are shown in Table 4 and Table 5.

Columns P1 and P2 were subjected to a test load, among other things, to verify the correct operation of the measuring devices installed inside the columns. Load tests were also conducted to obtain the load–settlement relationship for the columns, which is necessary for the proper design of the building’s foundation. A force was applied to the column heads, with the value increasing in ten stages. After the fifth stage, the columns were unloaded.

After the fifth stage, the reason for the unloading is to evaluate elastic recovery, calibrating measuring devices.

Regarding the results shown in Figure 21, during the mid-test unloading, the settlement of P1 is recovered similarly to P2, whereas the same does not occur at the end of the test.

When comparing the final deformation results recorded in columns P1 and P2, they are of the same order, i.e., approximately 50 microstrains. A different behavior was observed in Figure 21—after removing the load, there was no recovery of vertical deformation measured at the head of column P1. To clarify the matter, the results of all six load tests were compared. For columns numbered 50 (in axis D4/20), 126 (in axis D2/14), 329 (P2), 255 (P1), 434 (in axis D8/21), and 411 (in axis B8/16), the recovery of vertical deformation (the difference in settlement between the last load and unload step) is as follows: −4.60 mm, −3.82 mm, −6.71 mm, −0.46 mm, −5.06 mm, and −3.99 mm, respectively.

Clearly, the measurement in column P1 differs from the other five, which indicates a high probability of an error in reading the vertical deformation.

In column P1, the distribution of the axial force corresponded to typical column operation. Specifically, the estimated force values at all stages of the test load were highest for the sensor located at the highest point (P1_3), where the observed values were close to the force applied to the head at each stage of the load. The lowest force values were recorded by the sensor at the deepest point (P1_1). This response indicates good cooperation between the column side and the ground, as well as good quality of the column shaft.

When comparing these conclusions with the results of the CPT probing, it is important to note that the soil surrounding the P1 and P2 columns, at the location of the P1_2 and P2_2 sensors, exhibits significantly different cone resistance qc. For column P1, the resistance is approximately twice as high as that for column P2 (over 5 MPa for column P1 compared to 2.5 MPa for column P2, on average). The higher force values at the mid-height of column P1, compared to the upper part after the fifth stage and unloading, may also suggest variable values of the modulus of elasticity of the hardened concrete along the column.

Despite the differing behavior of the side surfaces and bases of columns P1 and P2, the load–settlement relationship determined for their heads is very similar (Figure 21). This indicates that, from the perspective of the structure’s overall behavior, the columns perform almost identically within the analyzed load range.

The conclusions from the axial force distribution interpretation along the shafts of columns P1 and P2 for different load steps can be compared to the findings resulting from the load–settlement relationship for their heads (Figure 21). After the final unloading for P1, practically no change in the head elevation was observed, while the head of column P2 moved upwards after the load was removed. The share of plastic deformation of the soil along and under column P2 is relatively small. Column P2 deforms elastically without the cooperation of the side surface (lack of lateral constraints). In the case of column P1, the side surface works much better, which is manifested by a practically complete reduction in the column head displacement upwards because the side surface counteracts the column from rising. 

The maximum force applied to the heads of columns P1 and P2 of 1200 kN causes an estimated elastic shortening of column P1 by about 8.3 mm (assuming the modulus of elasticity for the concrete column *E* = 21.7 GPa, diameter *D* = 0.40 m and length *L* = 18.8 m) and of column P2 by about 7.0 mm (assuming the same value of the modulus and diameter and length *L* = 15.8 m). The values estimated in this way are close to the values of the settlements of the heads of columns P1 and P2 measured during the test load (Figure 21). The estimated elastic shortening of column P1 observed on the load–settlement graph, taking into account good ground conditions around the column, may indicate the mobilization of friction along the entire length of the column, which results in a decrease in the axial force in the column. As shown in Figure 21, column P2 shortens elastically by about 6.0 mm. Permanent deformations of the substrate amount to about 2.0 mm. The extension (with excellent approximation) of the last linear section of the axial force dependence over depth for column P1 (Figure 22a) shows that the side surface transfers almost the entire force value applied to the column head, with practically no work performed by the column base. It follows from Figure 22b that different substrate–column interaction processes accompany the work performed by column P2. The agreement of the estimated and observed elastic deformations confirms the above interpretation of the axial force variation with depth, where the side layer does not cooperate with the ground in the first eight meters, both under load and during unloading. The return of elastic deformation of the column shaft is noticeable in the absence of resistance of the side layer. The work of the lower part of the column, for a depth greater than 8.0 m, causes the mobilization of friction at the column–ground interface and/or a slight settlement of the base, resulting in plastic deformation of the ground.

### 3.3. The Influence of Temperature Changes on the Measurement of Column Deformations

Sensors Pi_3 were installed at the point of the greatest temperature and atmospheric pressure fluctuation. For example, Figure 23 shows a graph of temperature changes as a function of time measured by sensor P1_3, placed in column P1. The graph shows that the temperature of the concrete near the head changes significantly and directly affects the values of deformations and, therefore, the estimated values of the axial force in the column.

Analyzing the actual deformations of concrete in a simplified way, we can distinguish free deformations, deformations resulting in the formation of stresses, and creep and shrinkage deformations. Neglecting rheological phenomena due to the relatively short measurement period, we can assume that the actual deformations (L) are the sum of deformations caused by the impact of temperature and deformations resulting from mechanical impacts, which generate stresses in whole or in part. In other words, these are such deformations that could be determined at the measuring point based on the measurement of the change in the base length made using an external measuring device, e.g., a caliper. Stress strains are theoretical strains that, using Hooke’s law, correspond to the values of stresses at the measuring point. Mechanical and non-mechanical interactions can cause them. For example, in a wholly restrained beam element, its length does not change due to temperature, and therefore, no strains occur. Using the temperature measurement, however, we can determine the theoretical value of the strains that would occur if the element were free to change length. Temperature changes in a restrained beam generate stresses that do not cause strains.

In the analyzed case, based on measurements with wire sensors, the actual strains ε(L) were determined, and then stress strains ε(σ) were separated from them (Figure 23). In the further part of the work, only the changes in stress strains ε(σ) over time will be analyzed.

### 3.4. Distribution of Force on the Elements of the Combined Piled Raft Foundation

The stress (pressure) change in the subsoil under the foundation slab was analyzed as a function of time (Figure 24). As mentioned earlier, the string pressure transducers were located in such a way as to enable the determination of the part of the foundation slab through which the load is transferred to the subsoil from the point support of the column type. One of the stress sensors was installed in a depression in the vicinity of the measured concrete column (S8), while the other two were in the middle of the spans, between the adjacent depressions (S7), and between the depression and the subsequent columns (S9)—compare. The measurements were designed to provide the information necessary to determine the vertical load distribution between the subsoil under the foundation slab and the concrete columns.

The automatic measurement system was launched on 20 May 2016 at 16% of the construction progress about the shell of the entire building (compare Table 1), i.e., after the foundation slab and the ceiling above story—two were completed in the part between axes C1-C5/15′-22′ and after concreting the foundation slab between axes B12-C7/10-15 (in the measurement area). In connection with the above, a correction was introduced to the stress measurement, consisting of taking into account the initial stress resulting from the thickness of the slab, i.e., 30 kPa for sensor S8 in the recess and 20 kPa for the two remaining sensors S7 and S9.

Based on the stress measurements in the subsoil under the foundation slab outside the recess (sensors S7 and S9), assuming for analysis the stress value from measurements taken between day 400 and 500, i.e., approx. 40 kPa, it can be hypothesized that the soil, apart from the foundation slab’s dead weight, transfers the load from the aboveground part of the structure at a level of approximately 20 kPa. The area of the slab within the column, where the recess of the foundation slab with an area of 3.5 m × 3.5 m is located, transfers to the ground a load that is more than twice as high, i.e., the stresses measured in the ground are on the order of 80 kPa, of which 30 kPa is the dead weight of the foundation slab, and 50 kPa is the load transferred by the column to the foundation.

To determine the load distribution on the subgrade and concrete columns, a comparison was made of the stress values transferred to the subgrade by the aboveground part of the structure (Figure 24) and the forces mobilized in the concrete columns (from Figure 25) with the total value of the force transferred by the column to the foundation slab. This force was estimated by the structural designer at the level of *Fc*,*d* = 10,125 kN based on the analysis of the FEM model of the aboveground part of the structure. Assuming the average load factor at the level of 1.2 (static and strength calculations were carried out based on the PN-B standard package), the average value of the force in the column can be estimated as Fc,k = 8438 kN. The additional load on the ground beyond the slab depth, which was measured as 20 kPa in the area of 7.75 m × 10.30 m − 3.5 m × 3.5 m = 67.6 m^2^, allows us to estimate the value of the force taken up by the ground under the slab at the level of Fr2,k = 1352 kN. The additional load on the ground under the slab recess, determined based on the measured stress of 50 kPa in the area of 3.5 × 3.5 = 12.3 m^2^, gives the force value Fr1,k = 613 kN. In total, the plate transfers a force of *Fr*,*k* = 1965 kN to the subgrade. The value of the axial force estimated after 788 days based on the measurement of concrete deformations with the P1_3 sensor located near the head of the P1 column is *Fp*,*k* = 869 kN. Considering the actual distribution of seven columns within the recess (Figure 25), it can be assumed that the total load taken by the group of columns is Fpg7,k = 7 × 869 kN = 6083 kN. The sum of the values of the force taken by the ground Fr,k = 1965 kN and the columns Fpg7,k = 6083 kN is Fc’,k = 8048 kN is very close to the force value specified by the structural designer, i.e., 8438 kN. The difference is about 4.8%. Finally, it can be assumed that in the analyzed plate–pile system, the subsoil takes over Fr,k/Fc’,k = 1965/8048 = 24.4% of the total load transferred to the foundation from the column and the Fpg7,k/Fc’,k column = 6083/8048 = 75.6% of this load. The lack of results observable between days 520 and 560 of monitoring is due to a break in data readings caused by the service provider, which resulted from the expiration of the initial monitoring service agreement. Once a new contract was signed, the readings were resumed. This is a limitation of the study. P1 to P6 is the column load with respective sensors to measure the displacement.

### 3.5. Analysis of the Behavior of Columns as a CPRF Element

Figure 25 shows the changes in the values of axial forces in columns P1 and P2 as a function of time, estimated based on deformation measurements using sensors placed along the tested columns.

After about 100 days of observation in column P2, sensor P2_3, which is located in the vicinity of the column head, no correct readings were observed.

During the measurements of deformations in column P4, 120 days after their commencement, the lack of correct readings was found in two sensors: the lowest (P4_1) and the highest located (P4_3) sensor. The fault caused by the broken wiring was removed in sensor P4_3 after 560 days.

During the installation of the measuring system in column P6, the repeated lifting and pressing of the steel I-beam into the concrete mix damaged the cable of the lowest-located sensor P6_1.

After approximately 70 days of readings in column P6, the middle sensor of column P6_2 was not receiving correct readings. The sensor was restored to working order after 560 days.

For all columns, after the conventional date of completion of the shell stage of the structure, i.e., after 201 days from the start of automatic measurements, a uniform increase in deformations was observed in all sensors installed in a given column.

A different issue is the absence of results for sensors numbered P2/3 and P6/1, which were damaged, for example, during installation. Yet, another issue concerns the lack of readings in sensors numbered P4/3 and P3/2, where it should be noted that due to excessive stretching or shortening of the string in the sensor—i.e., going beyond the measurement range—the sensors did not provide valid data within the expected range of strain/force. Such results were deliberately removed from the force vs. time chart. Once the readings returned to the valid measurement range, the data were again included in the chart.

Generally, it can be assessed that in column P3, friction along the side surface is mobilized the least. The force diagrams in the upper part of the column show similar values, while they overlap in the lower part. Column P1 behaves similarly, but the decrease in axial force with depth is more pronounced here, which may indicate better mobilization of friction. Column P2 shows better cooperation with the soil medium.

Columns P5, P6, and P4 observe the highest values of the estimated axial forces, which indicates that these columns operate very effectively.

The analysis of the change in the axial force value as a function of time (Figure 25) over 788 days allowed for the following conclusions to be drawn:

Column P1 (No. 257)

Column P1 works in a group of five or seven columns if we consider the additional two anchor columns made to carry out the test (Figure 26) at a spacing of 1.4 m (approx. 3.5D). Considering the anchor columns, column P1 works as a middle column.

Figure 27 compares the distribution of the axial force along column P1 for the last reading taken into account in this analysis, i.e., after 788 days from the start of measurements on 20 May 2016, with the change in the value of the axial force from the test load of the column. The course of the axial force along the column for the last reading leads to the conclusion that the side surface of the column in the plate–pile system works more weakly than in the single-layer system. The value of the axial force along the length of the column in the plate–pile system decreases slightly, i.e., by approx. 230 kN over the measured length of 10 m, in contrast to a single column, where the decrease in the force value along the length is apparent. The reduction in the contribution of the CPRF column sidewall to the load transfer is compensated by better mobilization of force in the lower unmeasured part of the column with a length of 8.8 m, including the base. This can be explained by increased friction on the sidewall in the lower part of the column and/or increased mobilization of the soil resistance under the column base.

The observations led to the conclusion that the statement about the existence of the so-called “dead zone” that typically occurs at small axle spacing between columns of 3D should be considered correct. This situation can be explained by the limitation of friction mobilization on the side due to the reduction in the relative displacement of the upper section of the column with respect to the surrounding soil. As a result of the pressure of the slab on the columns and the ground, the columns and the surrounding soil under the slab are subject to identical displacement. This interpretation is consistent with the conclusions presented in regarding the interactions between piles and the interactions between piles and the slab, according to which the friction on the side of the pile/column in the combined piled raft foundation is mobilized only in its lower part, i.e., on 1/3 to ¼ of the total pile length, i.e., 6.3 m in the case of our tests. In their work [31,32] indicated that the shaft friction distributions they presented for small center-to-center spacings, i.e., 3D, suggest that for a small range of settlements (0.03D), the raft in a piled-raft foundation negatively influences the distribution of unit shaft friction along the pile in the zone directly beneath the raft (up to 6 m, or 1/5 L for a central pile, and up to 25 m, or 0.83L for corner and edge piles). For greater settlements (0.10D), the behavior of the piles in a group and in the piled-raft foundation is very similar, although the performance of the central pile in the piled-raft foundation is noticeably better. The reduction in shaft friction mobilization directly beneath the raft results from the lack of relative displacement between the pile shaft and the surrounding soil.

Column P2 (No. 329)

The P2 column works in a five-column arrangement (after considering the additional two anchor columns), with a spacing of 2.30 m (approx. 5.85D).

The P2 column should be defined as the middle one in the column arrangement. Considering the column spacing close to 6D, according to the conclusions from the numerical analysis [10], it should be expected that the P2 column will work similarly to a single column.

Later in the work, a more extensive analysis for column P2 was performed, supplementing the field measurements with the results of numerical analysis.

Column P3 (No. 222)

The P3 column works in a three-column arrangement (Figure 26) as a middle column with a spacing of 1.43 m (3.6D).

As stated above, in column P3, friction along the side surface is mobilized the least in comparison to other columns. Very similar force values in three sensors, practically throughout the entire measurement period, indicate that the column load of 750 kN is transferred to a small extent by the first measured section of the column (approx. 130 kN) and, above all, by the lower, unmeasured part of the column, i.e., the side surface together with the base. Relating these observations to Kazenbach’s conclusions [7] from the monitoring of piles under the Commerzbank Tower building in Frankfurt, it can be assumed that columns embedded in load-bearing sands underlying much weaker dust mobilize a high value of friction on the side surface.

Columns P4, P5 and P6

It is assumed that columns P4, P5 and P6 operate in similar ground conditions and under similar load conditions. Column P5 operates in a single-column configuration, column P6 in a two-column configuration (spacing 2.83 m), and column P4 in a three-column configuration (spacing 2.33 m), as shown in Figure 28.

From the direct readings, the full distribution of force values along the length of column P5 can be estimated. In columns P4 and P6, although time-limited, reading is possible at the middle sensor (P4_2 and P6_2) and at the column head (P4_3 and P6_3).

Comparing the readings from the highest positioned sensors (Pi_3) in all three columns, it can be concluded that the force values on the last day of measurement are very similar to each other (1227 kN, 1203 kN, and 1148 kN for columns P4, P5, and P6, respectively).

Comparing the axial force drop on the upper measured section of the three columns, between sensors Pi_3 and Pi_2, it can be observed that the column side surface P4 works best, then P6 and P5, with the axial force decreasing by 595 kN, 362 kN and 243 kN, respectively.

To sum up, based on the distribution of estimated axial force values along the length of the column on the last day of measurement, based on the change of the axial force value with depth, it can be concluded that the behavior of columns P1, P2 and P3 is similar to the behavior of typical middle (inner) piles in a plate–pile system. Columns P1, P2 and P3 are located inside the group of columns. The small differences in the interpreted force values indicate poor side surface performance. The reduction in the side surface friction observed at the top of the columns indicates the existence of a “dead zone”.

Based on the measurements carried out, the behavior of columns P4, P5 and P6 can, with high probability, be compared to the behavior of typical columns’ corners/edges in the spacing of the order of 6D. Columns P4, P5 and P6 were located on the edge of the column group. The mobilization of friction on the side surface along the entire length of the column results in a significant decrease in the axial force along the column.

### 3.6. Detailed Analysis of the P2 Column Based on Field Measurements and FEM Calculations (ZSoil)

Based on the measurement of the actual deformations of concrete columns, a part was distinguished as stress, and then the axial force values were derived. This approach was particularly important for the sensors closest to the heads, where the most significant temperature changes were observed. In the case of columns P2 and P4, it was impossible to measure the strain in the column just below the foundation slab after 100 and 120 days, respectively.

Hence, to determine the course of the change of the axial force in the column head during P2, as a supplement to the readings from the P2_3 sensor during the field tests, a numerical analysis was carried out using the finite element method in the ZSoil program. The analysis also verified the possibility of using alternative tools to determine the values of axial forces in the column caused only by changes in load without the influence of temperature. The column choice resulted from possibly calibrating the model based on the load–settlement relationship from the test load. Moreover, thanks to the geodetic monitoring of the foundation slab settlements on the benchmark located directly above the P2 column, it was possible to obtain the information needed to compare the work of the combined piled raft foundation column with the column working independently.

#### 3.6.1. FEM Model

The FEM model [33,34] was limited to a detailed analysis of the part of the structure located between the B12-C7/10-15 axes (red contour) with column P2, taking into account the influence of the part located between the C1-C5/15′-22′ axes and the parts of the neighboring buildings with uniformly distributed load shown in Figure 29. The model considered the point average load from the most unfavorable combination of loads from all stories of the analyzed building acting on the foundation slab (from ARSA to Zsoil). The load in the form of concentrated forces was applied to the foundation slab at the locations of columns and walls. The model reproduced the actual arrangement of columns with their lengths from the as-built documentation. The columns were modeled as piles (beams) with a D = 0.40 m diameter. To stiffen the structure, all stories were taken into account in the FEM model. 

In order to model the subsoil, the constitutive soil model HSs (hardening soil model with small stiffness [35]) was used, for which the parameters were selected based on the available results of field and laboratory tests (Table 6).

The parameters adopted for the constitutive model of the subsoil medium were verified by comparing the load–settlement relationship for column P2 during the test load (Figure 30) and the settlement of the foundation slab located between axes B12-C7/10-15 of the building part (Figure 31).

Comparing the results of field tests and numerical simulations, a relatively good convergence with the results of the test load of column P2 can be observed (Figure 30). The numerically estimated settlement of the foundation slab compared to the geodetic measurements indicates a two- or three-fold overestimation of the results (Figure 31A,C), assuming the full load of the building. Assuming that 60% of the target weight of the building is applied to the foundation slab, the subsoil settles quite similarly to reality (Figure 31A,B). It is worth noting that the geodetic measurement on 12 June 2018 did not show any differences (except for the R11_H benchmark, where the change was 1 mm) compared to the measurement from 30 January 2017, i.e., from the 254th day from the start of automatic measurements. Based on the assumptions made at the beginning of the introduction, on the 201st day from the launch of the measuring platform, the progress of work on the building structure was approximately 50%.

#### 3.6.2. Analysis of Results

The analysis of the changes in the value of the axial force as a function of time for the measured columns showed that this course is bilinear. In the conventional time of 200 days, the increase in the axial force for all correctly functioning sensors is more significant than for later measurements. Geodetic measurements of the foundation slab settlements over time confirmed these observations.

For deformation measurements taken after day 201, i.e., for the finishing phase of the building, taking into account the estimated change in the force values determined based on the measurements with individual sensors (P2_2 and P2_1), it was estimated that the finishing works, which translated into a further increase in load, progressed at a rate of 0.04% per day. Hence, for further analysis, the steps of 55%, 60%, 70% and 80% of the total load were conventionally assumed as 325, 450, 700, and 949 days from the start of automatic measurements on 20 May 2016.

The FEM analysis in ZSoil was performed assuming the nodal load from ARSA, as a result in the form of reactions from rigid supports, on which the entire structure of the building was based, including the foundation slab. This situation resulted in the inability in ZSoil to take into account the direct transfer of the load from the foundation slab’s own weight to the subsoil. Therefore, the values of axial forces in the numerically modeled columns were corrected by the value of the foundation slab’s weight.

Considering that automatic measurements began after the foundation slab and the ceiling above the “−2” level were completed, the modification of axial forces also included the ceiling weight. In total, according to Table 1, the values of axial forces, estimated numerically at each load step, were reduced by the value of forces generated for 16% of the full load. In accordance with the records from the construction journal regarding the commencement of concreting of individual levels of the building, for each characteristic load step, the corrected value of axial force at the depth corresponding to the location of sensors P2_3, P2_2, and P2_1 was read from the ZSoil program and compared with the force values from the column deformation measurements (Figure 32). Additionally, the values of axial force at a depth of 14.48 m estimated in ZSoil were plotted with the label ZSoil P2/0.

Comparative analysis of the results indicates a good convergence of the force values in sensors P2_2 and P2_1, especially in the last three steps of the construction implementation phase and at the building completion stage. After exceeding the conventional step of 70% of the building load, the force values from field measurements stabilized, while the forces from the numerical analysis continued to increase. Similarly to the observations from the building settlements, which stabilized and did not change the settlements estimated numerically for the 60% load step, a further increase in the axial force resulting from the FEM analysis can be considered unjustified.

This is most likely because the load values from the most unfavorable combination were used in the numerical analysis and were overestimated compared to the actual values. The numerically determined value of force as a function of time in the P2_3 sensor, located in the vicinity of the column head, allowed for a comparison of the behavior of a single column with the behavior of a column of a combined piled raft foundation. Figure 33 shows the course of the load settlement relationship of the column head. The actual measurement of the foundation slab settlement in the benchmark located directly above the P2 column was compared with the value of force in the P2_3 sensor read from the FEM model. Scholars monitor structures using field monitoring systems and finite element methods [36,37,38]. These methods help to minimize structural risks in buildings and large bridges [39,40,41].

The hyperbolic curve approximated of actual measurements of the test (dashed line). CPRF force vs. plate settlement and column working in CPRF show force readings from FEM analysis after axial force correction vs. plate settlement above column P2 and results from FEM analysis with axial force correction (ZSoil—P2_3 CPRF).

Based on the comparison of the load–settlement (Ns) curve for a single column from the test load with the Ns curve for a column with a combined piled raft foundation, it can be concluded that the CPRF column, in the range of small deformations of the soil medium (correspondingly, the settlement of the column head at the level of several millimeters), is characterized by lower stiffness, i.e., it settles more than an independent column at a similar load value.

Considering the ongoing development of construction works, determining the force–time course in the P2_3 sensor, located near the column head and at an additional depth of 14.48 m, made it possible to supplement the actual measurements and compare the axial force distribution along the length of the combined piled raft foundation column with the force distribution measured in a single column during test loads (Figure 34).

Based on the linear approximation of the numerically estimated values in ZSoil, the force values in sensors P2_2, P2_1, and P2_0 were determined for the force value read in sensor P2_3 from the test load. The compiled information made it possible to draw the following conclusions:For the first three load steps (137 kN, 274 kN, 412 kN), the value of the axial force in the FPP column is reduced to a lesser extent compared to the single column, which can be interpreted as a weaker mobilization of friction on the side surface of the column.For subsequent load steps, we observe different behavior of the CPRF column—a more significant decrease in the axial force along the length of the CPRF column indicates a more significant mobilization of friction on the side surface than in the case of a single column.The involvement of the side surface in the cooperation in the ground with relatively low mechanical parameters in the CPRF sub-plate zone, compared to the lack of mobilization of friction on the side surface of the column working independently, can be explained by the increase in vertical normal stresses in the soil medium around the column, caused by the pressure of the foundation slab.

## 4. Conclusions

The measurement system, consisting of a setup for monitoring stress changes in the soil beneath the building’s slab, as well as a system for measuring deformation of concrete columns, was presented. It can be considered a significant contribution to the development of knowledge regarding the interaction between the elements of a combined piled–raft foundation.

In the analysis, actual deformations resulting from temperature changes were separated from stress-induced deformations arising from changes in load, and only stress-induced deformation changes over time were further interpreted.

Using measurements of soil stresses and forces in the columns, the distribution of load between the subsoil and the concrete columns was estimated, indicating that in the analyzed piled-raft system, the columns bear 75% of the total load from the pillar. The analysis of changes in axial force values in six columns over time showed that after exceeding the conventional date marking the completion of the raw structural state, i.e., approximately 201 days after the start of automatic measurements, the increase in force in all sensors for a selected column occurred proportionally. Among the examined columns, the following behaviors were observed:-Typical behavior for middle columns with small changes in axial force measured along the column, indicating weak lateral skin friction and simultaneously suggesting the existence of a “dead zone”.-Typical behavior for corner/boundary columns with spacing on the order of 6D, with the ability to mobilize friction along the entire length of the column, ultimately resulting in a distinct decrease in axial force at depth.

Based on finite element analysis (FEA) in ZSoil, missing readings at the head and base of one of the columns were supplemented, while simultaneously comparing these readings with those from other correctly functioning sensors. Comparing the behavior of an individual column with the behavior of the piled-raft foundation column, it was deduced that the piled-raft column, within the range of small settlements (several millimeters), exhibits lower “stiffness” than a single column. However, with larger settlements (0.06D), an increase in load-bearing capacity and stiffness of the piled-raft foundation column was observed, in line with conclusions from archival analyses.

## Figures and Tables

**Figure 1 sensors-25-03460-f001:**
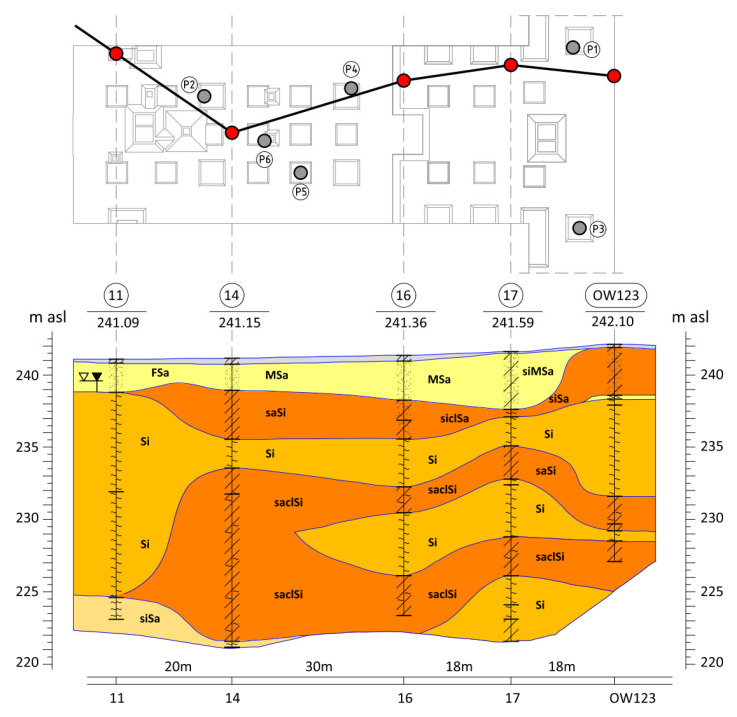
Geological engineering cross-section of the ground beneath the analyzed building. The location of the tested concrete columns (P1 to P6) was marked [14].

**Figure 2 sensors-25-03460-f002:**
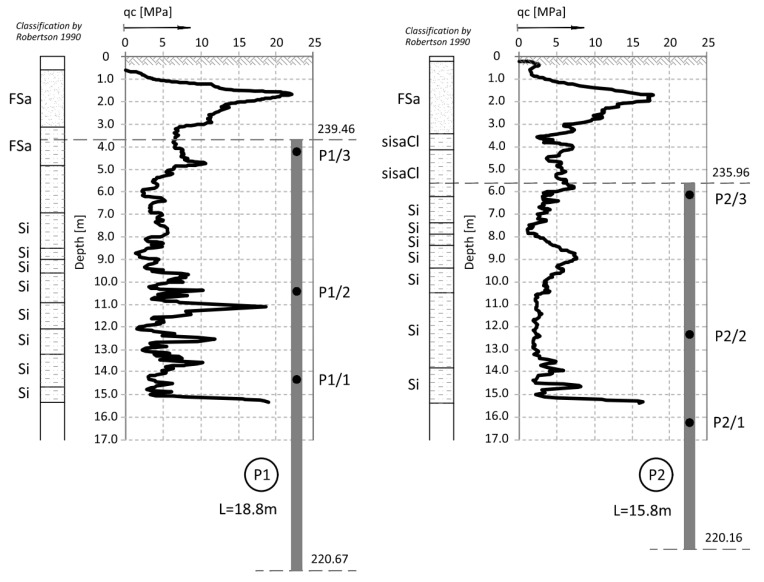
CPT probing in the vicinity of columns P1 and P2 [15].

**Figure 3 sensors-25-03460-f003:**
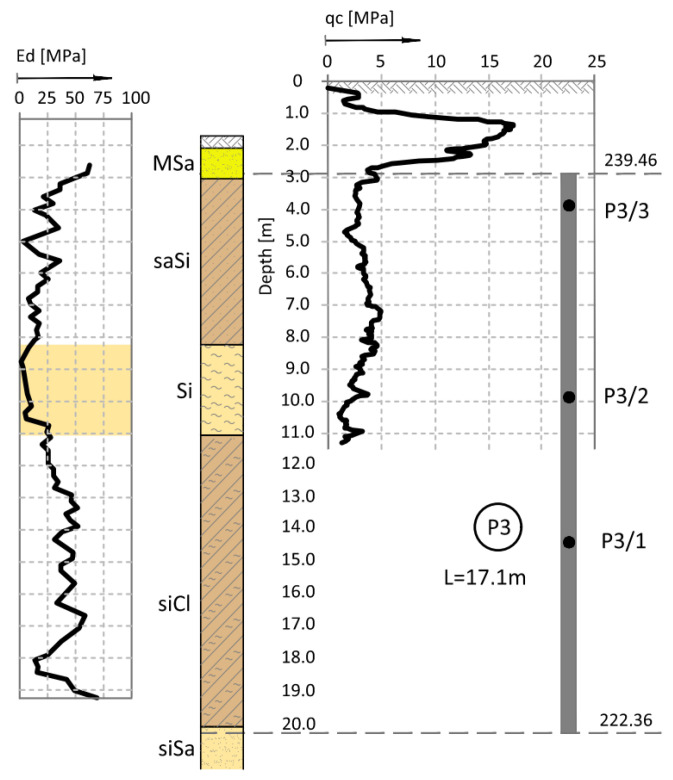
DMT probing (Ed module value) and CPT probing in the vicinity of column P3 [14,15].

**Figure 4 sensors-25-03460-f004:**
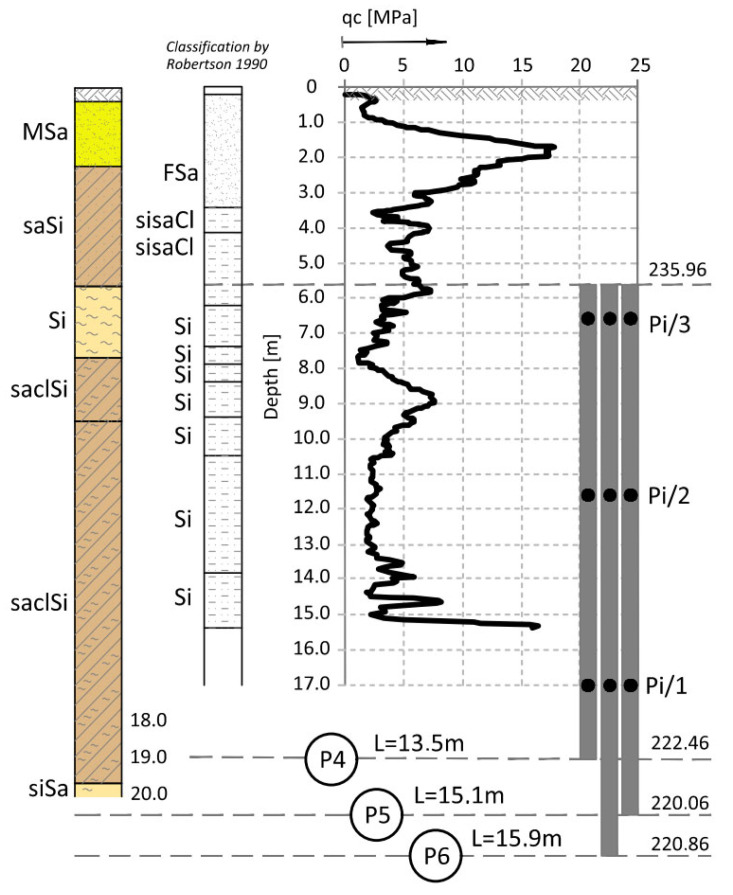
CPT probing in the vicinity of columns P4, P5, and P6 [15].

**Figure 5 sensors-25-03460-f005:**
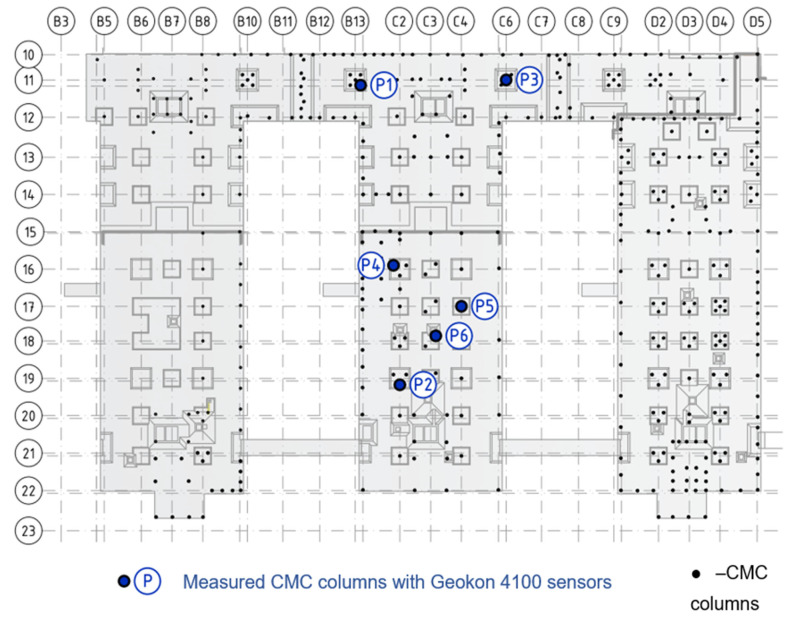
Location and marking of measured concrete columns (P1–P6).

**Figure 6 sensors-25-03460-f006:**
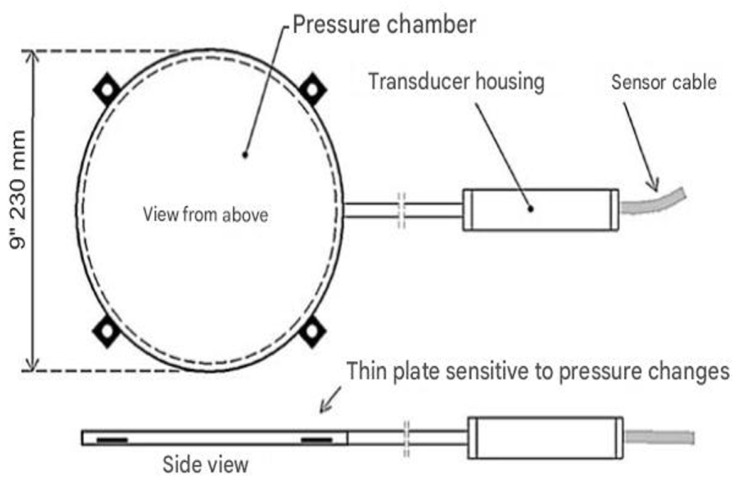
Construction of a string pressure transducer.

**Figure 7 sensors-25-03460-f007:**
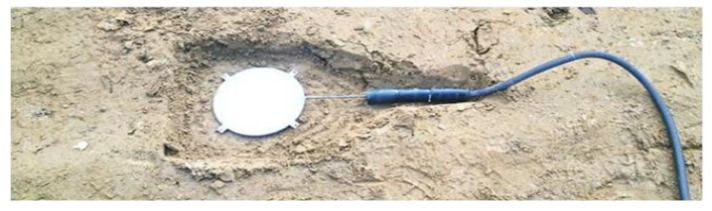
View of the recess prepared for the installation of the pressure transducer.

**Figure 8 sensors-25-03460-f008:**
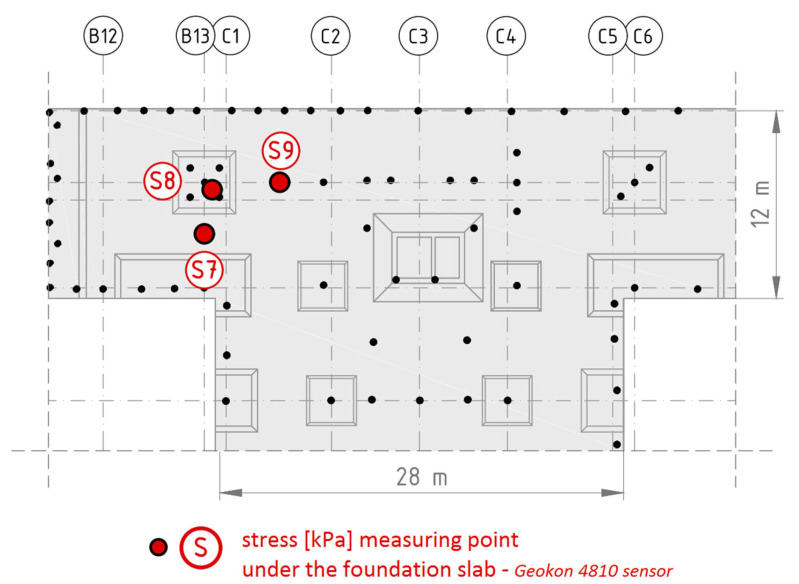
Location and names of stress measurement points in the ground under the foundation slab.

**Figure 9 sensors-25-03460-f009:**
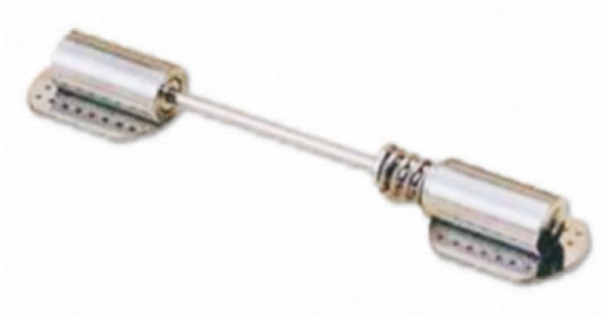
Geokon 4100 string strain sensor.

**Figure 10 sensors-25-03460-f010:**
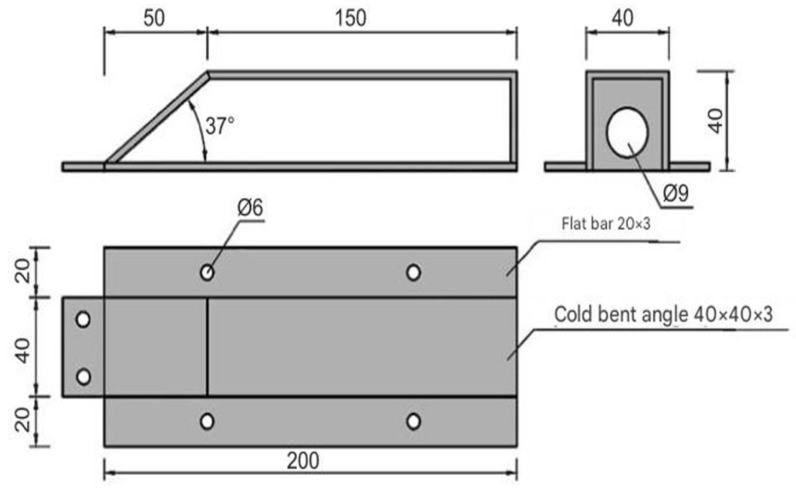
Construction of protective casings.

**Figure 11 sensors-25-03460-f011:**
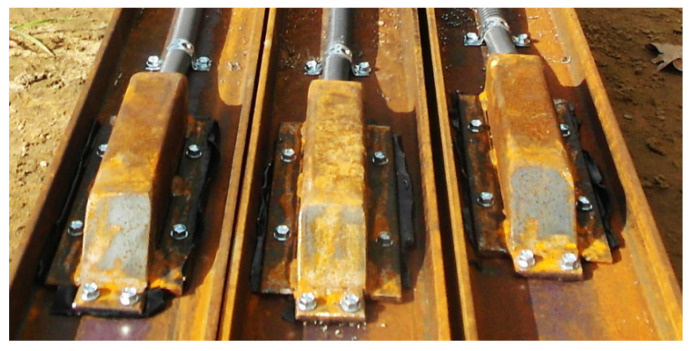
Protective casings installed on IPE120 profiles.

**Figure 12 sensors-25-03460-f012:**
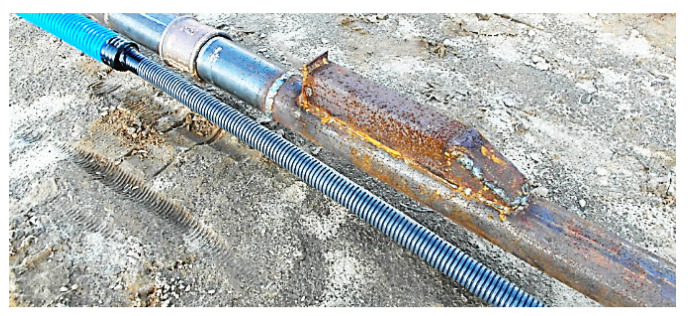
Protective casings installed on RO pipes 2″× 2.9.

**Figure 13 sensors-25-03460-f013:**
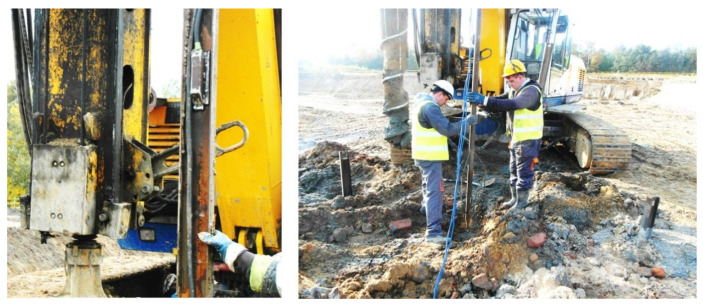
View of steel profiles with sensors during the insertion into the concrete mix.

**Figure 14 sensors-25-03460-f014:**
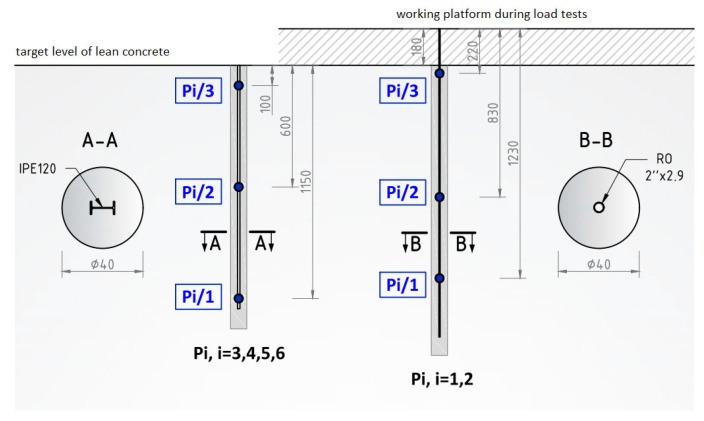
The method of installing deformation sensors and the adopted numbering.

**Figure 15 sensors-25-03460-f015:**
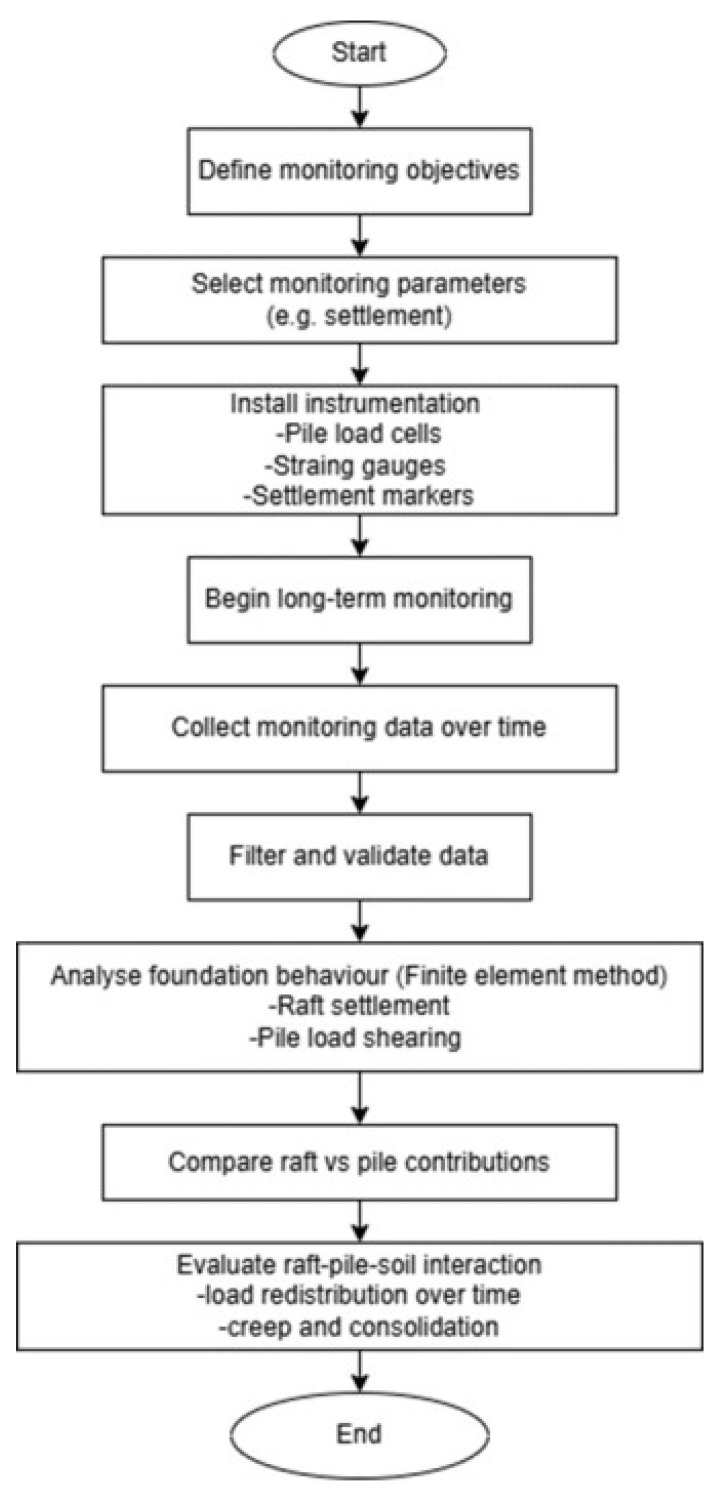
Flowchart of the study.

**Figure 16 sensors-25-03460-f016:**
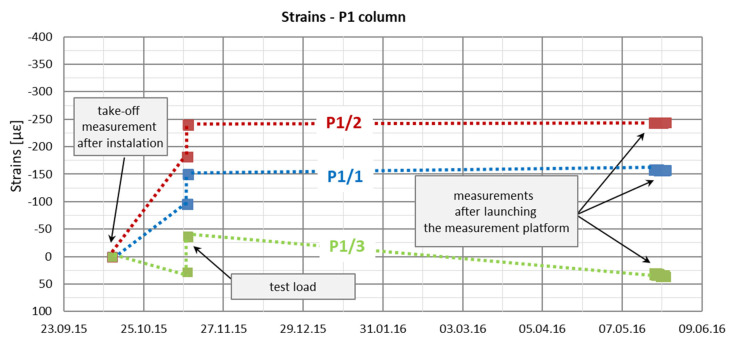
Strain graphs of the P1 column in the period from the take-off measurements to the start of automatic measurements.

**Figure 17 sensors-25-03460-f017:**
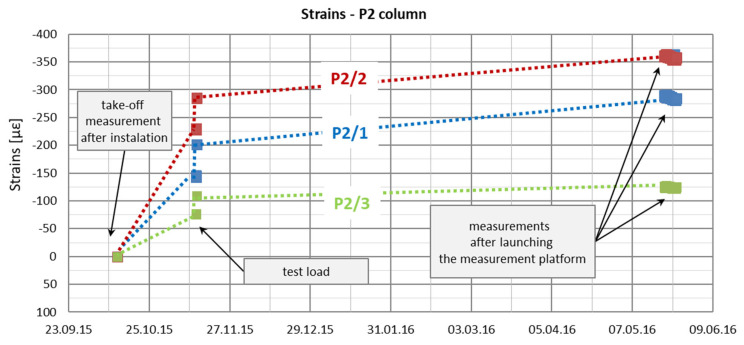
Strain graphs of the P2 column in the period from the take-off measurements to the start of automatic measurements.

**Figure 18 sensors-25-03460-f018:**
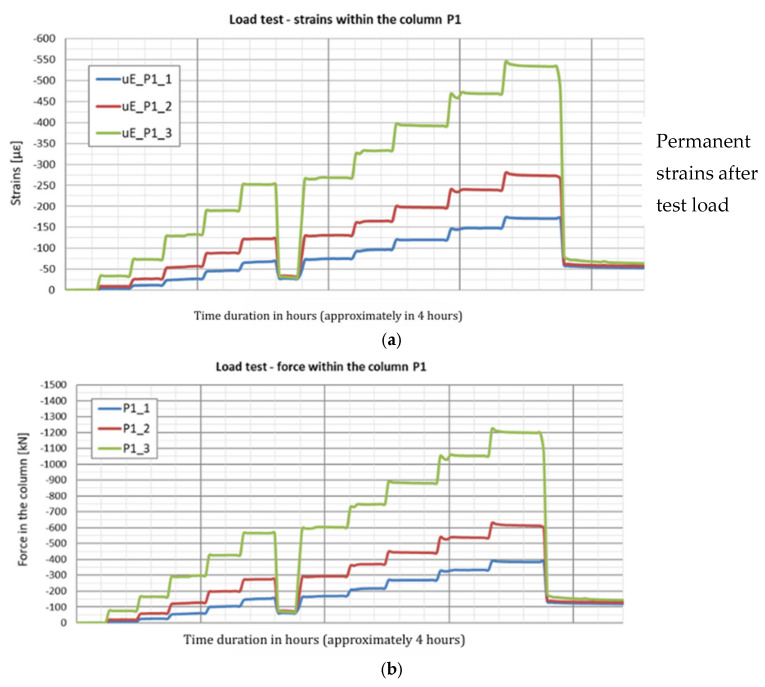
(**a**) Deformation of column P1 and (**b**) axial force in column P1 during the load test.

**Figure 19 sensors-25-03460-f019:**
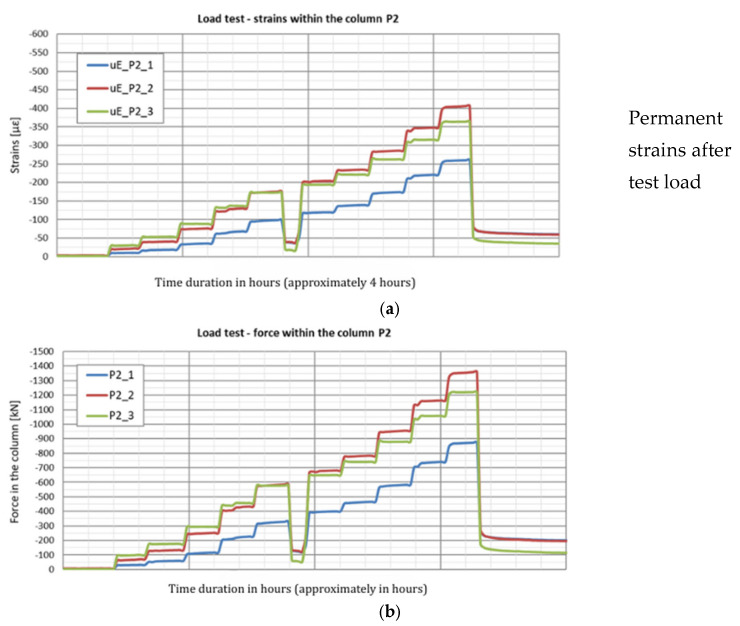
(**a**) Deformation of column P2 and (**b**) axial force in column P2 during the load test.

**Figure 20 sensors-25-03460-f020:**
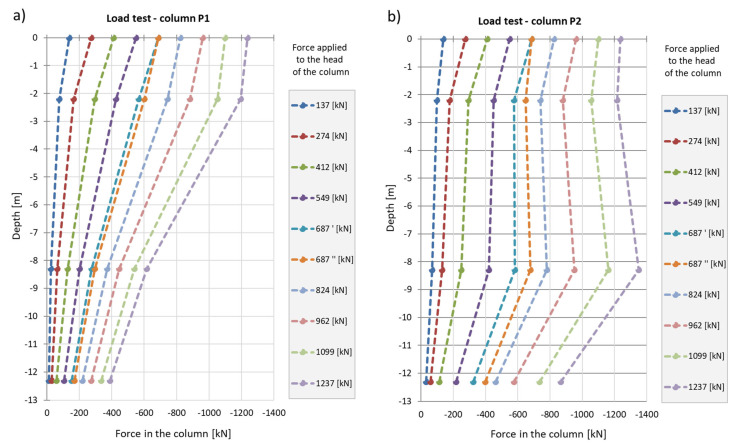
Graphs of the force distribution along columns (**a**) P1 and (**b**) P2 during the test load. Graphs were made based on the strain readings.

**Figure 21 sensors-25-03460-f021:**
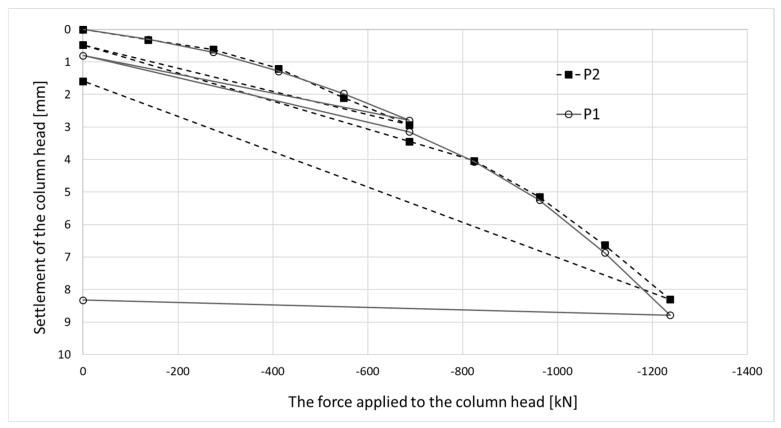
The load–settlement relationship for the head for columns P1 and P2 (load test results).

**Figure 22 sensors-25-03460-f022:**
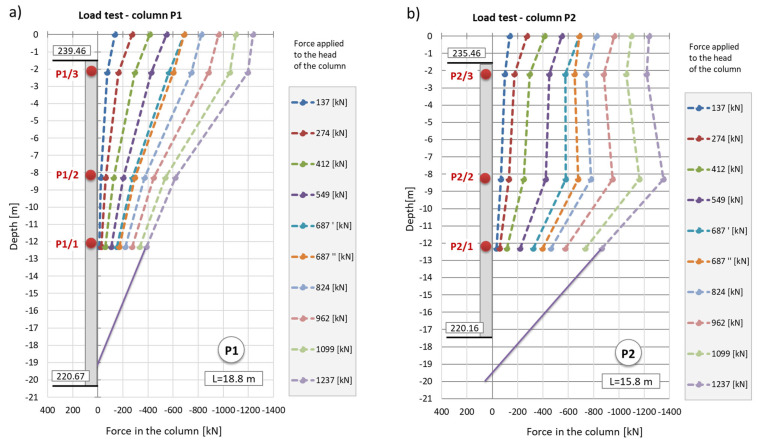
Hypothetical extension of the force distribution diagram along columns (**a**) P1 and (**b**) P2 during the test load.

**Figure 23 sensors-25-03460-f023:**
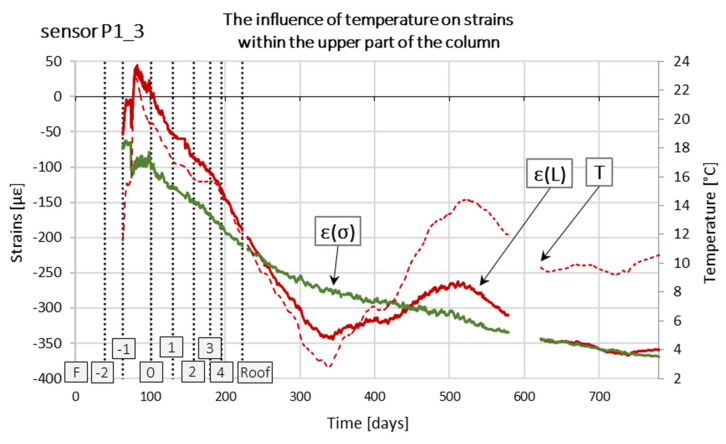
Influence of temperature on the determined strain values. Designations: ε(L)—real strains; ε(σ)—stress strains.

**Figure 24 sensors-25-03460-f024:**
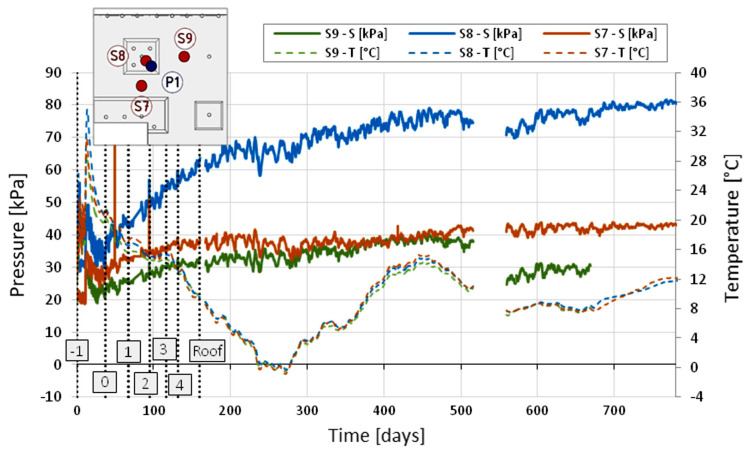
Change in the stress value in the ground under the foundation slab as a function of time for selected locations, taking into account the correction resulting from the stresses existing in the ground at the moment of starting the measurement.

**Figure 25 sensors-25-03460-f025:**
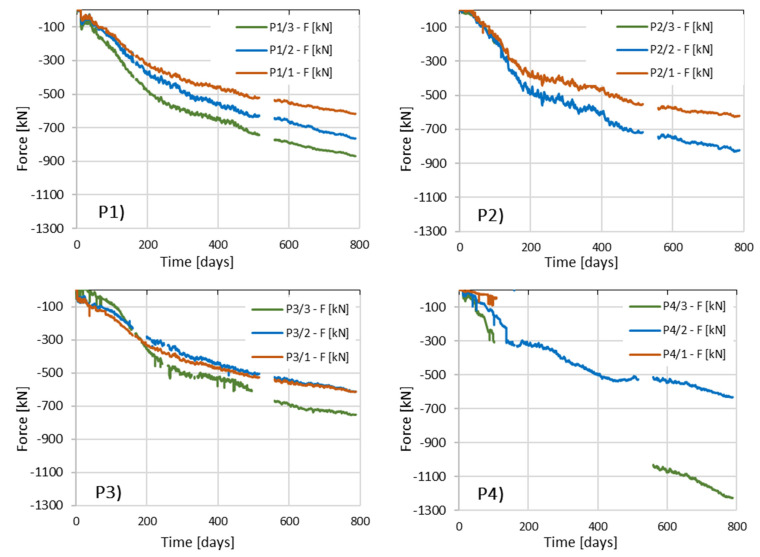
Change in the estimated value of the axial force as a function of time based on the measurement of concrete deformations.

**Figure 26 sensors-25-03460-f026:**
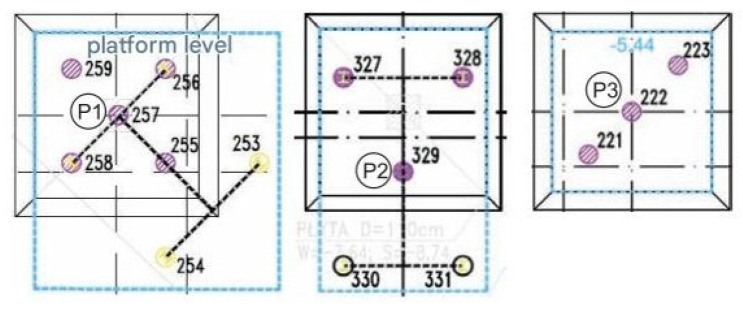
Column arrangement around columns P1, P2, and P3.

**Figure 27 sensors-25-03460-f027:**
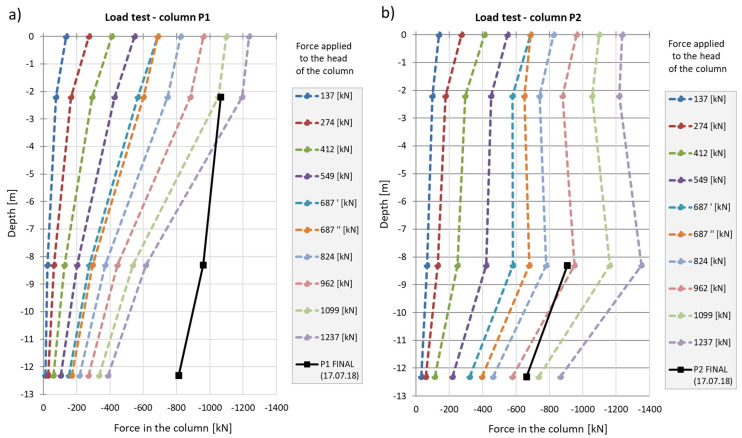
Distribution of force along columns (**a**) P1 and (**b**) P2 during the test load and for the last reading during the construction of the building (solid black line).

**Figure 28 sensors-25-03460-f028:**
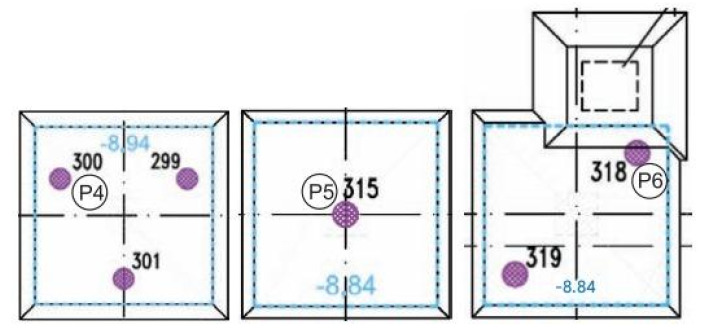
Arrangement of columns around columns P4 (No. 300), P5 (No. 315), and P6 (No. 318).

**Figure 29 sensors-25-03460-f029:**
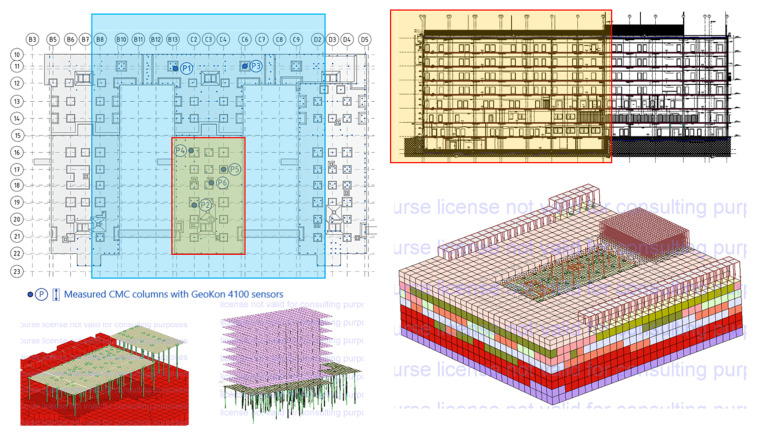
FEM model (ZSoil) of the selected part of the building.

**Figure 30 sensors-25-03460-f030:**
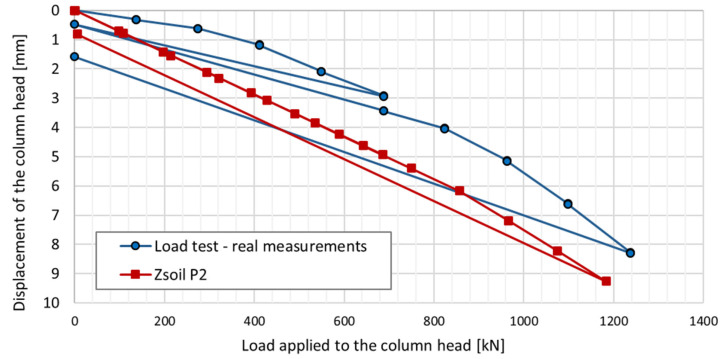
Load–settlement relationship of the P2 column head. Comparison of actual measurements with FEM simulation (ZSoil P2).

**Figure 31 sensors-25-03460-f031:**
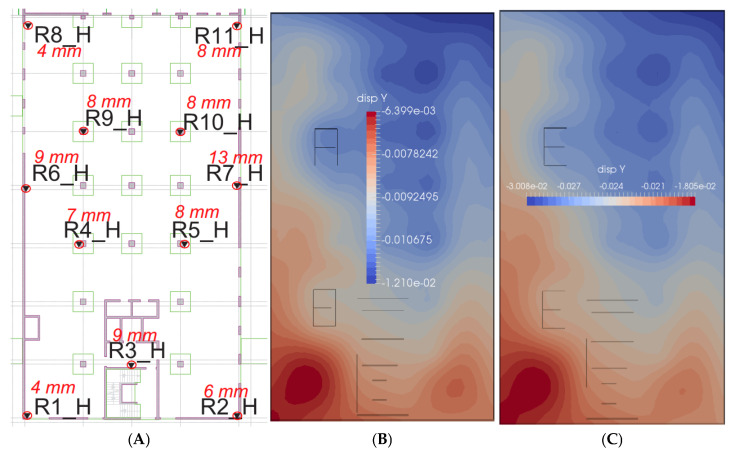
Comparison of the foundation slab settlements based on the results of (**A**) geodetic monitoring (last measurement from 12 June 2018, identical as on 30 January 2017) and FEM analysis (ZSoil) (**B**) for 60% load and (**C**) for 100% load.

**Figure 32 sensors-25-03460-f032:**
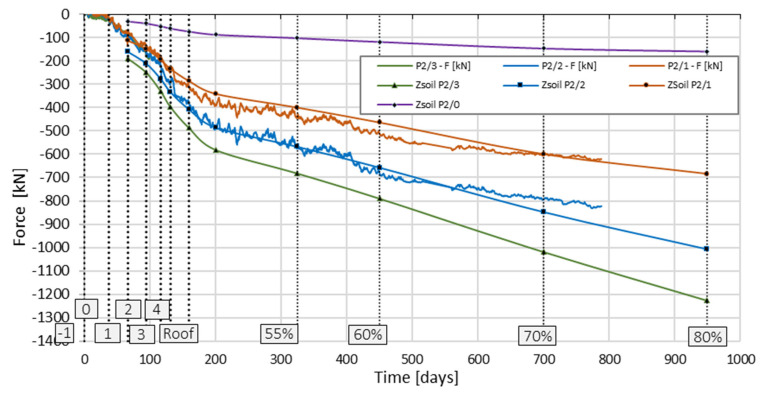
Change in axial force values as a function of time in column P2 for three depths. Force values were estimated from field tests and read after correction from the FEM model (ZSoil).

**Figure 33 sensors-25-03460-f033:**
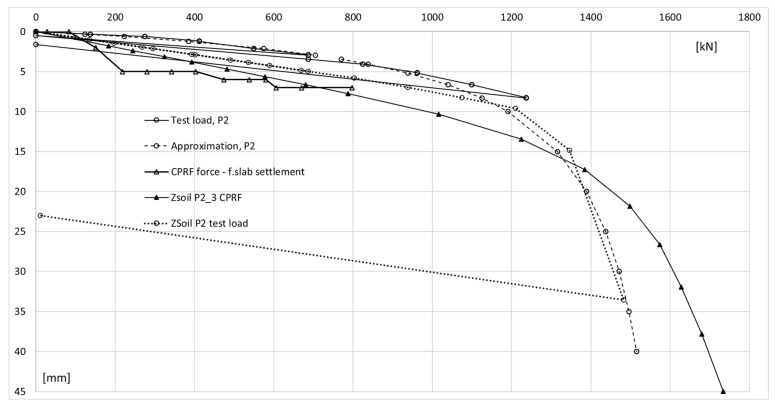
Load–settlement relationship of the P2 column head (for a column working alone vs. for a column in a slab–pile foundation). Actual measurements from the column test (continuous line, empty circle marker); FEM simulation of the test (ZSoil Test Load P2, dotted line).

**Figure 34 sensors-25-03460-f034:**
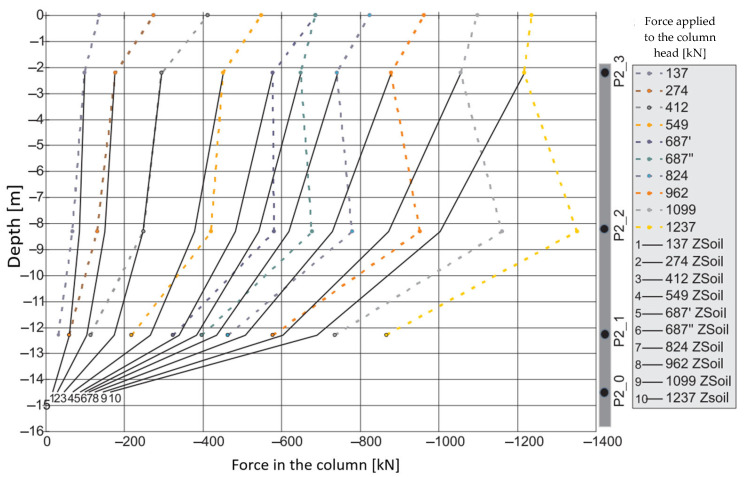
Distribution of force along column P2 during test load (actual results) and during the construction of the building based on the FEM analysis (ZSoil).

**Table 1 sensors-25-03460-t001:** Concreting schedule by the construction journal records and an indicative layout of the floors in the analyzed building.

**Time [Days]**	**Start of Concreting**	**Time Between Stages**	**% of Total Load**	**Stage**	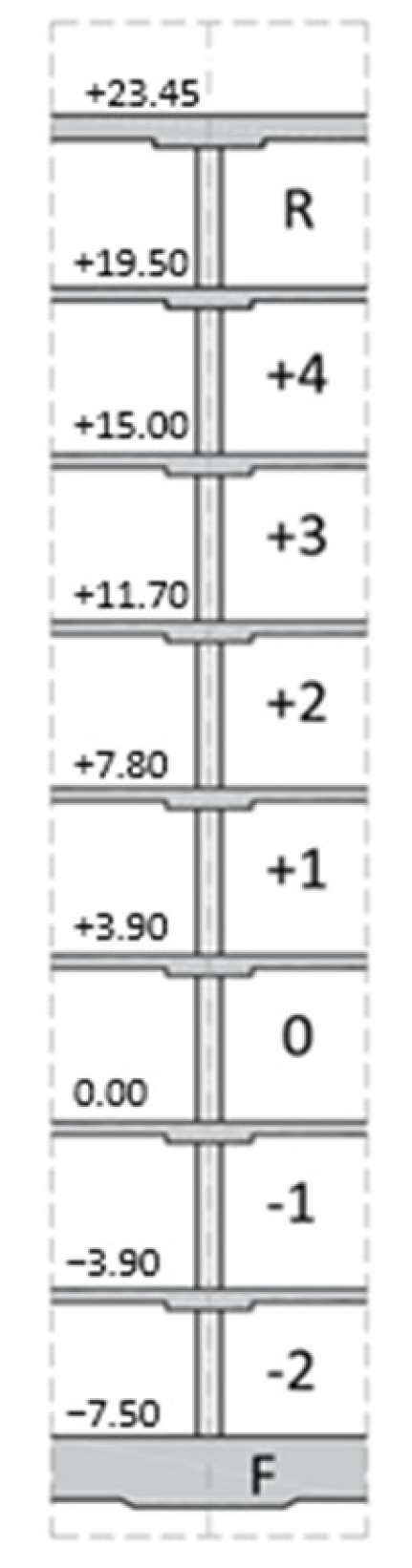
	18 March 2016	0	0.00	Foundation slab	
	24 April 2016	39	0.11	–2	
0	20 May 2016	24	0.16	Kick-off automatic measurements	
0	20 May 2016	0	0.16	–1	
38	27 June 2016	38	0.21	0	
67	26 July 2016	29	0.26	+1	
95	23 August 2016	28	0.31	+2	
117	14 September 2016	22	0.35	+3	
132	29 October 2016	15	0.40	+4	
160	27 October 2016	28	0.45	Roof (R)	
201	24 November 2016	41	0.50	Contractual completion of the shell stage	

**Table 2 sensors-25-03460-t002:** Technical parameters of the Geokon 4800 pressure sensor.

Transducer Type	String
Measurement range	170 kPa
Overload	150% range
Resolution	±0.025% of the range
Accuracy	±0.1% of the range
Linearity	<0.5% of the range
Chamber dimensions	6 mm × 230 mm
Transducer dimensions	150 mm × 25 mm
Material	Stainless steel
Temperature range	−20 to +80 °C

**Table 3 sensors-25-03460-t003:** Technical parameters of the Geokon 4100 strain sensor.

Sensor type	String
Measurement range	9000 µε
Resolution	0.4 µε
Accuracy	±0.1% of range
Linearity	<0.5% of range
Temperature range	20 to +80 °C
Active sensor length	51 mm

**Table 4 sensors-25-03460-t004:** Load test results for P1.

**Stage**	**Force (kN)**	**Pressure (MPa)**	**Time (min)**	**Readings (mm)**		**Average (mm)**	**Increment (ds) (mm)**		**Settlement (mm)**	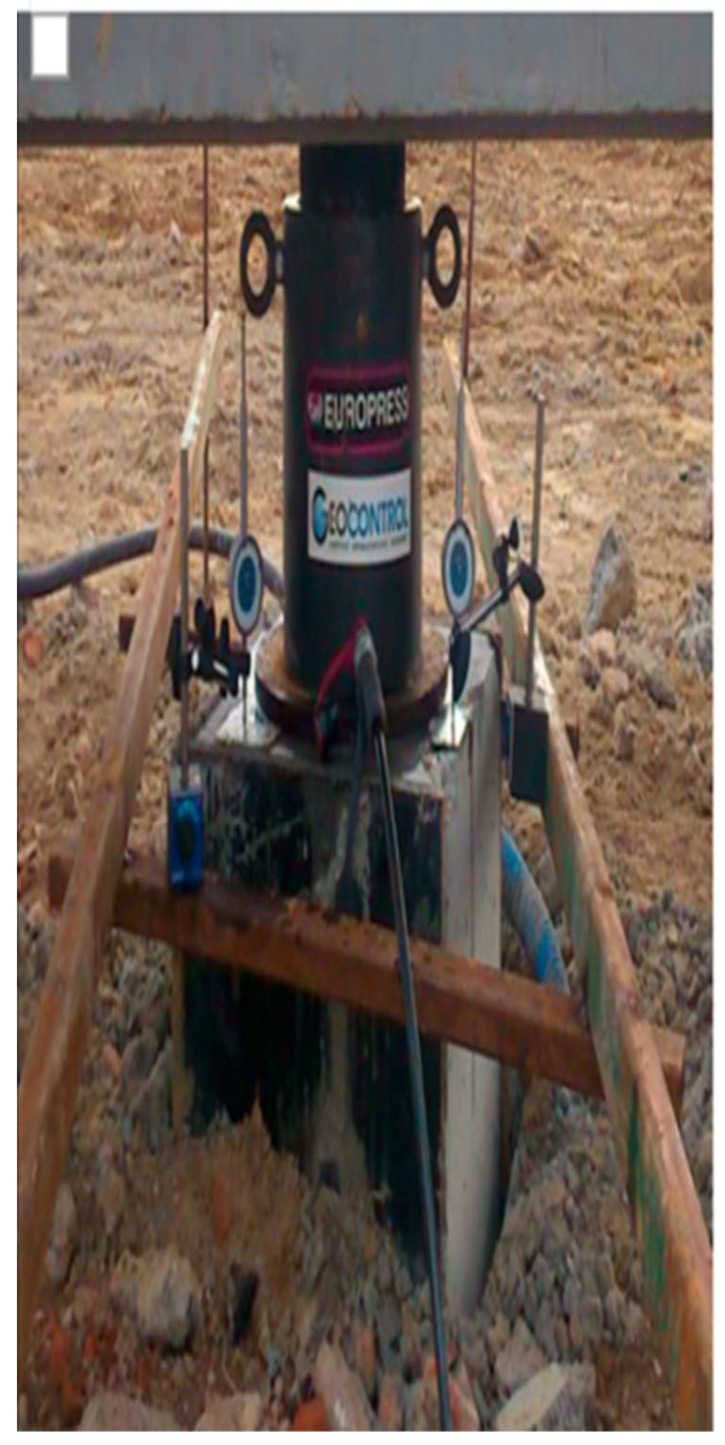
0	0.00	0.0	---	45.00	45.00	45.00	45.00	0.00	--	
1	137	50.0	10	44.70	44.75	44.72	44.72	0.28	0.28	
2	274	50.0	10	44.67	44.73	44.69	44.70	0.03	0.03	
3	412	100.0	10	44.30	44.38	44.25	44.31	0.39	0.69	
4	274	100.0	10	44.29	44.37	44.24	44.30	0.01	0.70	
5	412	150.0	10	43.73	43.87	43.62	43.74	0.56	1.26	
6	412	150.0	10	43.70	43.85	43.69	43.71	0.03	1.29	
7	549	200.0	10	43.00	43.26	42.86	43.04	0.67	1.96	
8	687	200.0	10	42.99	43.26	42.84	43.03	0.01	1.97	
9	687	250.0	10	42.13	42.55	42.00	42.23	0.80	2.77	
10	687	250.0	10	42.11	42.54	41.98	42.21	0.02	2.79	
11	0	0.0	10	44.18	44.33	44.08	44.20	−1.99	0.80	
12	687	250.0	10	41.86	42.35	41.70	41.97	2.23	3.03	
13	687	250.0	10	41.76	42.26	41.60	41.87	0.09	3.13	
14	687	250.0	10	41.75	42.24	41.58	41.86	0.02	3.14	
15	824	300.0	10	40.76	41.76	40.63	40.95	0.91	4.05	
16	824	300.0	10	40.76	41.45	40.60	40.94	0.01	4.06	
17	962	350.0	10	39.56	40.56	39.48	39.87	1.07	5.13	
18	962	350.0	10	39.46	40.46	39.38	39.77	0.10	5.23	
19	962	350.0	10	39.46	40.45	39.35	39.75	0.01	5.25	
20	1099	400.0	10	37.80	39.22	37.72	38.25	1.51	6.75	
21	1099	400.0	10	37.72	39.18	37.65	38.18	0.06	6.82	
22	1099	400.0	10	37.67	39.13	37.60	38.13	0.05	6.87	
23	1237	450.0	10	35.77	37.80	35.58	36.38	1.75	8.62	
24	1237	450.0	10	35.64	37.69	35.44	36.26	0.13	8.74	
25	1237	450.0	10	35.59	37.65	35.39	36.21	0.05	8.79	
26	0	0.0	10	36.30	38.77	34.95	36.67	−0.46	8.33	

**Table 5 sensors-25-03460-t005:** Load test results for P2.

**Stages**	**Force (kN)**	**Pressure (MPa)**	**Time (min)**	**Readings (mm)**		**Average (mm)**	**Increment (ds) (mm)**		**Settlement (mm)**	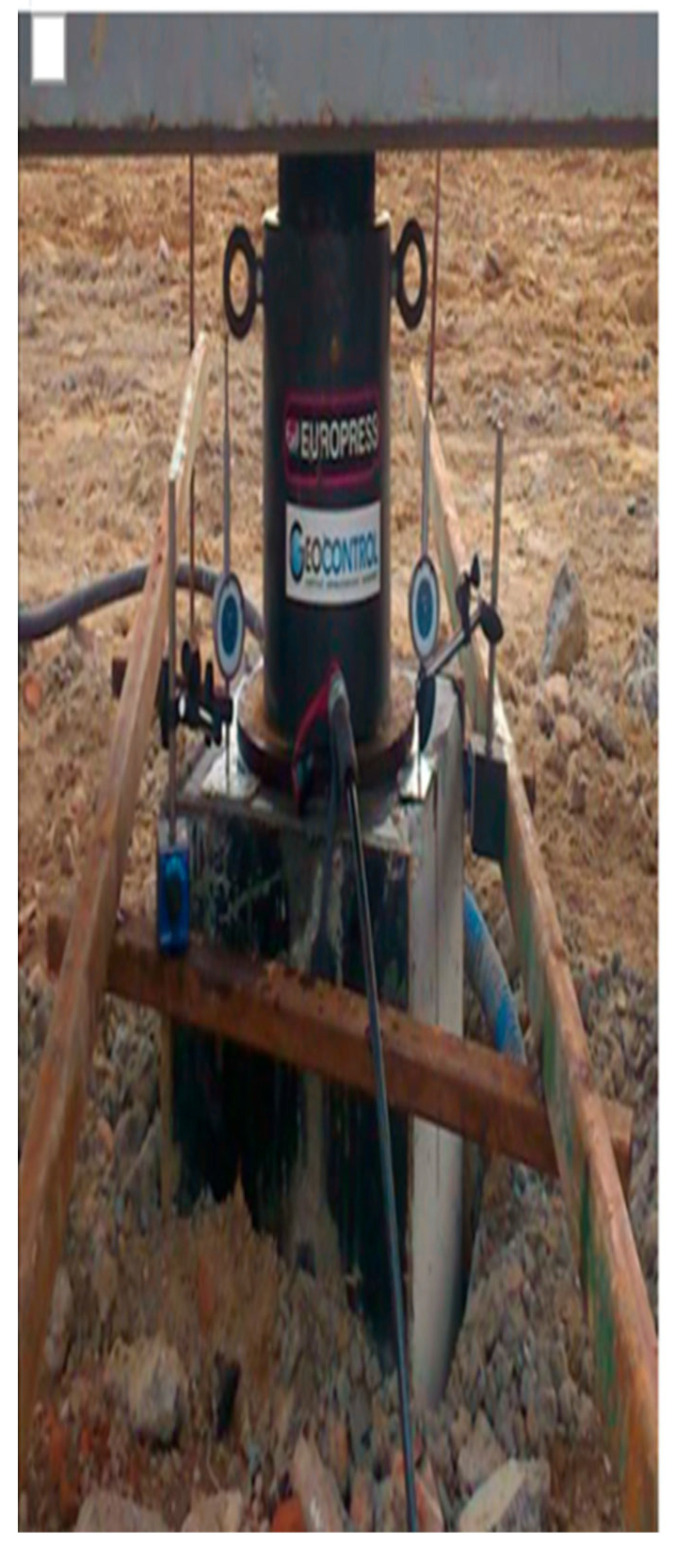
0	0.00	0.0	---	35.00	35.00	35.00	35.00	0.00	---	
1	137	50.0	10	34.73	34.68	34.65	34.69	0.31	0.31	
2	137	50.0	10	34.72	34.67	34.65	34.68	0.01	0.32	
3	274	100.0	10	34.39	34.37	34.39	34.38	0.30	0.62	
4	274	100.0	10	34.39	34.37	34.39	33.38	0.00	0.62	
5	412	150.0	10	33.86	33.80	33.82	33.83	0.56	1.17	
6	412	150.0	10	33.83	33.78	33.79	32.80	0.03	1.20	
7	549	200.0	10	32.99	32.90	32.87	32.92	0.88	2.08	
8	549	200.0	10	32.99	32.88	33.83	32.90	0.02	2.10	
9	687	250.0	10	32.24	32.04	32.01	32.10	0.80	2.90	
10	687	250.0	10	32.21	32.02	31.98	32.07	0.03	2.93	
11	0	0.0	10	34.56	34.51	34.51	34.53	−2.46	0.47	
13	687	250.0	10	31.66	31.52	31.56	31.58	2.95	3.42	
14	687	250.0	10	31.64	31.50	31.54	31.56	0.02	3.44	
15	824	300.0	10	31.07	30.90	31.94	30.97	0.59	4.03	
16	824	300.0	10	31.06	30.89	30.93	30.96	0.01	4.04	
17	962	350.0	10	30.06	29.80	29.78	29.88	1.08	5.12	
18	962	350.0	10	30.03	29.76	29.75	29.85	0.03	5.15	
19	1099	400.0	10	28.67	28.30	28.20	28.39	1.46	6.61	
20	1099	400.0	10	28.64	28.29	28.18	28.37	0.02	6.63	
21	1237	450.0	10	27.43	26.01	26.74	26.73	1.64	8.27	
22	1237	450.0	10	27.40	25.99	26.71	26.70	0.03	8.30	
23	0	0.0	10	33.53	33.39	33.32	33.41	−6.71	1.59	

**Table 6 sensors-25-03460-t006:** Parameters of the HSs model of the subsoil.

	Soil Layer								
Parameter	IIb	IIIb	IVa	IVb	IVc	IVd	Va	VI	W
*E*_0_^*ref*^ (kPa)	408,000	276,000	312,000	120,000	252,000	336,000	168,000	516,000	500,000
*γ*_0.7_ (×10^4^)	1.61	2.28	1.63	1.68	2.14	1.82	4.50	1.43	1.43
*E*_*ur*_^*ref*^ (kPa)	90,000	60,000	75,000	20,000	50,000	70,000	32,000	110,000	110,000
*E*_50_^*ref*^ (kPa)	27,000	16,000	25,000	6000	13,000	22,000	10,000	35,000	35,000
*σ^ref^* (kPa)	80	99	86.8	59.4	104.5	110.5	135	112.5	100
*m* (-)	0.5	0.6	0.5	0.55	0.6	0.6	0.75	0.5	0.5
*v*_*ur*_ (-)	0.2	0.2	0.2	0.2	0.2	0.2	0.2	0.2	0.2
*R_f_* (-)	0.9	0.9	0.9	0.9	0.9	0.9	0.9	0.9	0.9
*C′* (kPa)	0	2	0	0.5	4	5	6.5	0	0
*φ′* (°)	39	35	36	32	34	36	23	41	41
*ψ* (°)	7	6	5	2	6	2	0	10	10
*e*_*max*_ (-)	0.85	1.2	0.85	2.2	1.4	1.2	2.0	0.85	0.85
*f_t_* (kPa)	0	0	0	0	0	0	0	0	0
*D* (-)	0.25	0.25	0.25	0.25	0.25	0.25	0.25	0.25	0.25
*E*_*oed*_^*ref*^ (kPa)	27,000	16,000	25,000	6000	13,000	70,000	10,000	35,000	35,000
*σ_oed_^ref^* (kPa)	216.2	232.4	266.3	126.3	237.0	268.3	221.0	331.0	294.1
*K*_0_*^NC^* (-)	0.37	0.426	0.41	0.47	0.44	0.41	0.61	0.34	0.34
*q*^*POP*^ (kPa)	200	200	200	200	200	200	200	200	30
*K*_0_*^SR^* (-)	0.37	0.426	0.41	0.47	0.44	0.41	0.61	0.34	0.34
*K*_0_ (-)	2	1.5	0.78	0.9	0.95	0.85	1.5	0.75	0.5

## Data Availability

The data are provided by the corresponding author.

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
