# Peer review of "Assessment of the Interaction of the Combined Piled Raft Foundation Elements Based on Long-Term Measurements"

_sensors, 2025, doi:10.3390/s25113460_

Round 1

Reviewer 1 Report

Comments and Suggestions for Authors

This paper evaluates the mutual influence between CPRF components. Changes in stress values in the subsoil, as well as strain measurements in the vertical direction of concrete columns, were recorded to assess the load distribution between the CPRF components. The content of the paper is relatively rich and the framework is reasonable. The paper still needs some modifications before it can be accepted and published. 1.The format of this paper seems to be different from others, and the innovation of the paper should be combined with the abstract. 2. As mentioned in the first point, the innovation of the paper abstract is not clear enough and lacks the effectiveness of indicator quantification methods. It is suggested to add some quantifiable indicators to showcase the results of the paper. 3. In terms of introduction, it is necessary to strengthen the description of research background and significance, and Engineering Structures can be added, 2025, 327: one hundred and nineteen thousand five hundred and twenty-three 4. The materials and methods section needs to be further refined to facilitate the reproducibility of the results. 5. A deeper introduction is needed for Figures 17 and 18. 6.What are the reasons for the missing data in Figures 22, 23, and 24. Is there any way to restore it? 7.From the perspective of the entire paper, it is recommended to add a flowchart to reflect the innovative points of the paper.

Author Response

Comments and Suggestions for Authors

This paper evaluates the mutual influence between CPRF components. Changes in stress values in the subsoil, as well as strain measurements in the vertical direction of concrete columns, were recorded to assess the load distribution between the CPRF components. The content of the paper is relatively rich and the framework is reasonable. The paper still needs some modifications before it can be accepted and published. 

Comments 1:

1.The format of this paper seems to be different from others, and the innovation of the paper should be combined with the abstract. 

Response 1:

This work was used to assess the long-term deformation analysis using field monitoring and finite element methods.

Comments 2:

  1. As mentioned in the first point, the innovation of the paper abstract is not clear enough and lacks the effectiveness of indicator quantification methods. It is suggested to add some quantifiable indicators to showcase the results of the paper. 

Response 2:

This work's original contribution is to validate the finite element analysis with field tests or monitoring of the deformation of the combined raft pile foundation. Field monitoring and FEA methods are used to compare the long-term deformation analysis and it helps to minimize time of monitoring.

Comments 3:

  1. In terms of introduction, it is necessary to strengthen the description of research background and significance, and Engineering Structures can be added, 2025, 327: one hundred and nineteen thousand five hundred and twenty-three.

Added this in the introduction to strengthen the study

Response 3:

Thank you for the recommendation. The reference was included reference number 31.

The monitored building is part of the newly constructed hospital in Kraków-Prokocim, which consists of a total of twenty facilities, including nine hospital buildings with a combined total area of 150,000 m² and a volume of 500,000 m³. The construction was completed within 4.5 years. The analyzed building was monitored for almost 800 days (see Fig. 31), which corresponds to approximately 80% of the total planned load (excluding variable load from crowds). The shell stage of the building was completed after approximately 200 days, reaching 50% of the total planned load (see Table 1).

Comments 4:

  1. The materials and methods section needs to be further refined to facilitate the reproducibility of the results.

Response 4:

Thank you for the recommendation. It is corrected and included in the section.

Comments 5:

  1. A deeper introduction is needed for Figures 17 and 18.

Response 5:

Thank you for the recommendation and suggestions. It is included in the section.

Comments 6:

  1. What are the reasons for the missing data in Figures 22, 23, and 24? Is there any way to restore it?

Response 6:

The lack of results observable between days 520 and 560 (Figures 24 and 25) of monitoring is due to a break in data readings caused by the service provider, which resulted from the expiration of the initial monitoring service agreement. Once a new contract was signed, the readings were resumed. This is the limitation of the study.

Comments 7:

 7.From the perspective of the entire paper, it is recommended to add a flowchart to reflect the innovative points of the paper.

Response 7:

Thank you for your comment. I included the flowchart.

Figure 1. Flowchart of the study

Reviewer 2 Report

Comments and Suggestions for Authors

In this study, long-term measurements of performance characteristics were performed to assess the mutual influence of the elements of Combined Piled Raft Foundation of a multi-storey frame monolithic building. Temperature and load deformations, as well as stress changes over time, were analyzed. Soil stresses and column deformations were recorded to assess the load distribution. The results of the numerical modeling of the foundation fragment were compared with field measurements to verify the model of interaction between the structure and the soil and to expand the test database.

The work is original and relevant to the construction industry, as the design of foundations with complex multilayer soils is a complex process. The work uses theoretical calculations based on the finite element method and large-scale tests that complement and correlate theoretical calculations. This adds credibility and gives confidence in the results.

This work significantly complements scientific research in this area, and its methodology can be applied to other complex projects.

The literature review covers mainly publications older than 5 years, which is a drawback.

The presentation of scientific material from tests to theoretical calculations is consistent and complete. There are no comments.

The conclusions are fully consistent with the question posed.

The graphs and tables give a complete picture of the set conditions and the results obtained.

In general, the work is interesting, with in-depth experimental and computational studies. The disadvantage is the small (10 out of 30) number of references to modern (up to 5 years old) literature sources.  

Author Response

Comments 1:

In this study, long-term measurements of performance characteristics were performed to assess the mutual influence of the elements of Combined Piled Raft Foundation of a multi-storey frame monolithic building. Temperature and load deformations, as well as stress changes over time, were analyzed. Soil stresses and column deformations were recorded to assess the load distribution. The results of the numerical modeling of the foundation fragment were compared with field measurements to verify the model of interaction between the structure and the soil and to expand the test database.

Response 1:

Comments 2:
The work is original and relevant to the construction industry, as the design of foundations with complex multilayer soils is a complex process. The work uses theoretical calculations based on the finite element method and large-scale tests that complement and correlate theoretical calculations. This adds credibility and gives confidence in the results.

Response 2:

Comments 3:

This work significantly complements scientific research in this area, and its methodology can be applied to other complex projects.

Comment 4:
The literature review covers mainly publications older than 5 years, which is a drawback.

Response 4:

Thank you for the recommendation and suggestion. I included the recent publications.

Comments 5:
The presentation of scientific material from tests to theoretical calculations is consistent and complete. There are no comments.

Comment 6:
The conclusions are fully consistent with the question posed.

Comment 8:
The graphs and tables give a complete picture of the set conditions and the results obtained.

Comments 9:
In general, the work is interesting, with in-depth experimental and computational studies. The disadvantage is the small (10 out of 30) number of references to modern (up to 5 years old) literature sources.  

Response 9:

Thank you for the comment and recommendations. It is included in the manuscript.

Reviewer 3 Report

Comments and Suggestions for Authors

This paper presents the results of an extensive experimental campaign conducted over a long period, aimed at analyzing the behavior of a building foundation consisting of a combined concrete slab and columns.

Six columns were instrumented with vibrating wire sensors, and three pressure transducers were installed.

The results were analyzed in detail using a finite element model.

The work is extensive and gathers a great deal of highly relevant information. While the results analysis is thorough, it is not exhaustive and could be further explored. For instance, aspects related to temperature compensation and the rheological effects of concrete in the piles should be noted.

With a view to improving the submitted document, some comments and suggestions follow.

In Chapter 2, there should be a better description of the building and a more detailed account of the construction sequence. Only towards the end of the article does it become clear that the construction may have continued for several hundred days (not quantified) after the concrete was poured for the roof slab.

Following the analysis sequence in Chapter 3 and considering the numbering of the sensors, it is suggested that in Chapter 2, subsections 2.3 and 2.4 be swapped. In fact, the more complete foundation plan shown in Figure 8 should appear before the excerpt shown in Figure 5.

Lines 180 and following – The references in subsection 2.3 appear to be misplaced. I suggest they be reviewed.

Line 312 and following – It would be important to have the temperature data at this first reading to allow for a correct interpretation of the data. It would also be important to understand whether the first reading captures the beginning of the heat of hydration or already the cooling phase of the concrete mass.

Line 329 and following – The deformation is divided into three periods. As written, it may suggest that most of the shrinkage occurred before the load test date (phase 1). I have difficulty to believe the load test date coincided with the change in the deformation slope. As the authors later conclude, since the zero point was not correctly established, “it is not possible to fully or accurately interpret the processes occurring in the column during the initial 30 days.”

Line 360 – I find it hard to understand how soil expansion would impose deformation on a pile (and sensor) so close to the surface, especially in a column that primarily works in end bearing. Might this deformation be instead related to the expansion of concrete in the presence of water (if applicable), or perhaps some bending in the column?

In Chapter 3, only the results for columns P1 and P2 (equipped with tubes) are presented, while results for columns P3 to P6 (with IPE sections) are omitted. Is there any reason for this selection? Were the other columns also tested? What was the selection criterion for the tested columns?

Line 362 – A description of the columns load test would be helpful: for instance, how the load was applied, loading values, test procedure, identified measured quantities, and the test setup. It becomes clear later that topographic measurements were used. What instruments were used, and what was their sensitivity?

Line 402 – In addition to the hypothesis raised by the authors regarding local cross-section reduction (leading to increased deformation), the possibility of eccentricity in the metal tube inside the cross-section should not be ruled out, as this could result in measuring possible pile bending.

The lack of recovery of the surface settlement of column P1 at the end of the load test (Figure 20) could be better explained. Part of this settlement (around half) can be attributed to the axial deformation observed in each of the piles (estimated at around 5 mm). At the end of unloading, the deformation recovery in both piles P1 and P2 is not very different (Fig. 17a and Fig. 18a). At least this amount of settlement should have been recovered similarly in both piles. If, as stated (line 432), “the side surface counteracts the column (P1) from rising,” then the axial deformation of the column should remain, which is not the case.

Moreover, regarding the results in Figure 20, no explanation is given as to why, during the mid-test unloading, the settlement of P1 is recovered similarly to P2, whereas the same does not occur at the end of the test. Was there a reference error in measuring the settlement of P1 at the end?

The sensor used (Geokon 4100) compensates for thermal effects when applied to steel. According to the manufacturer's technical specifications:

"If attached to a steel structure, the thermal coefficient of expansion of the steel vibrating wire inside the instrument is the same as that for the structure. This means that no temperature correction for the measured strain is required when calculating load-induced strains."

This raises a question because the deformation is imposed by the concrete, which has a slightly lower linear thermal expansion coefficient. This aspect is omitted in lines 473 and following. The strain correction likely takes into account this behavior of the sensor used. I suggest clarifying how the strains that generate stresses are derived in Figure 20.

Line 536 – The foundation slab description in Chapter 2 omits its plan dimensions. It is presumed that the 3.5 m × 3.5 m comes from somewhere, but it’s unclear where the 7.75 × 10.30 m dimension comes from.

Line 573 – In Figure 24, how do the authors justify the continuous increase in pile forces 600 days after construction? These forces were calculated from strain data. I estimate that the stresses imposed on the pile concrete approach 10 MPa. At this stress level, shouldn't a creep model for concrete be included? Couldn’t this apparent increase in forces be partly explained by concrete creep?

Still on this subject:

Line 747 – This comment regarding continued construction work and increasing column loads should appear earlier, when Figure 24 is presented.

Note that this increase in pile load is not accompanied by a similar increase in the load cell readings (see Figure 23), which again raises the question of whether the increase in forces in Figure 24 is real.

Line 597 – I suggest a better framing of the comparison being made between the “plate-pile system” and the “single-layer system.” I got a bit lost in the comparison.

Author Response

This paper presents the results of an extensive experimental campaign conducted over a long period, aimed at analyzing the behavior of a building foundation consisting of a combined concrete slab and columns.

Six columns were instrumented with vibrating wire sensors, and three pressure transducers were installed.

The results were analyzed in detail using a finite element model.

The work is extensive and gathers a great deal of highly relevant information. While the results analysis is thorough, it is not exhaustive and could be further explored. For instance, aspects related to temperature compensation and the rheological effects of concrete in the piles should be noted.

With a view to improving the submitted document, some comments and suggestions follow.

 Comments 1:

In Chapter 2, there should be a better description of the building and a more detailed account of the construction sequence. Only towards the end of the article does it become clear that the construction may have continued for several hundred days (not quantified) after the concrete was poured for the roof slab.

 Response 1:

The monitored building is part of the newly constructed hospital in Kraków-Prokocim, which consists of a total of twenty facilities, including nine hospital buildings with a combined total area of 150,000 m² and a volume of 500,000 m³. The construction was completed within 4.5 years. The analyzed building was monitored for almost 800 days (see Fig. 32), which corresponds to approximately 80% of the total planned load (excluding variable load from crowds). The shell stage of the building was completed after approximately 200 days, reaching 50% of the total planned load (see Table 1).

Comments 2:

Following the analysis sequence in Chapter 3 and considering the numbering of the sensors, it is suggested that in Chapter 2, subsections 2.3 and 2.4 be swapped. In fact, the more complete foundation plan shown in Figure 8 should appear before the excerpt shown in Figure 5.

 Response 2:

Thank you for your valuable recommendation. It is done according to your recommendation.

Comment 3:

Lines 180 and following – The references in subsection 2.3 appear to be misplaced. I suggest they be reviewed.

 Response 3:

Thank you for the suggestions. All citations are reviewed.

Comment 4:

Line 312 and following – It would be important to have the temperature data at this first reading to allow for a correct interpretation of the data. It would also be important to understand whether the first reading captures the beginning of the heat of hydration or already the cooling phase of the concrete mass.

 Response 4:

In line 337 – 340 is written : “In the first phase, column shortening was observed in all sensors except P1_3, most likely caused by temperature changes following the release of heat during cement hydration, along with concrete shrinkage. Shrinkage occurs due to structural changes in the cement paste caused by physicochemical processes of water loss during concrete setting and hardening.”

Temperature was measured at every step and was taken into account when determining e(s) - stress strains values, as part of the true stresses (line 462). It cannot be clearly determined whether the first measurement, taken on October 12, 2015, reflects the temperature increase due to the heat of hydration or the cooling phase of the concrete. Comparing the measured temperatures of 20.3 °C, 18.4 °C, and 20.7 °C recorded on October 12, 2015, in sensors P1/1, P1/2, and P1/3 respectively, with the values recorded a month later—11.2 °C, 11.4 °C, and 11.7 °C—it can be observed that the temperature of the concrete decreased along the entire length of the column. On May 20, 2016, the temperature stabilized across all sensors at 9.9 °C, 9.0 °C, and 11.6 °C, and remained exactly the same one year later.

In column P2, a temperature decrease over time was observed in all sensors. The recorded values for sensors P2/1, P2/2, and P2/3 were, respectively: 15.7 °C, 16.4 °C, and 15.5 °C on October 12, 2015; 10.2 °C, 10.2 °C, and 10.1 °C on November 12, 2015; and 9.2 °C, 8.9 °C, and 10.9 °C on May 20, 2016.

Finally, it should be concluded that the concrete temperature measurements indicate a cooling trend over time.

Comments 5:

Line 329 and following – The deformation is divided into three periods. As written, it may suggest that most of the shrinkage occurred before the load test date (phase 1). I have difficulty to believe the load test date coincided with the change in the deformation slope. As the authors later conclude, since the zero point was not correctly established, “it is not possible to fully or accurately interpret the processes occurring in the column during the initial 30 days.”

 Response 5:

Based on the observation of a temperature drop in concrete during the initial phase, it is logical to conclude that the cooling of concrete leads to its shrinkage (understood as a reduction in volume or length). A different behavior was observed in the upper part of column P1 – a noticeable elongation occurred in phase 1 and in phase 3.

The authors intended to indicate that applying the load during the trial loading of the columns led to a logical reduction in their length (all columns).

Comment 6:

Line 360 – I find it hard to understand how soil expansion would impose deformation on a pile (and sensor) so close to the surface, especially in a column that primarily works in end bearing. Might this deformation be instead related to the expansion of concrete in the presence of water (if applicable), or perhaps some bending in the column?

 Response 6:

Looking at the soil conditions near the P1 column head (location P1/3, Fig. 2), we can observe the presence of sand FSa, which creates a natural local reservoir for the accumulation of, for example, rainwater. On the other hand, knowing that sandstone aggregate was used in the concrete, one can point to the possibility of concrete swelling. Furthermore, attention should be drawn to the potential occurrence of alkali-silica reaction (ASR), which, in the presence of moisture, leads to the formation of a gel that absorbs water and causes expansion. Additionally, sandstone as an aggregate is characterized by high absorbability, which promotes water penetration into the concrete and may lead to its swelling.

However, as in the first phase, this can be attributed to the presence of water in the substrate, which causes soil swelling—a phenomenon observed exclusively in the upper part of the column.

Comments 7:

In Chapter 3, only the results for columns P1 and P2 (equipped with tubes) are presented, while results for columns P3 to P6 (with IPE sections) are omitted. Is there any reason for this selection? Were the other columns also tested? What was the selection criterion for the tested columns?

 Response 7:

In order to verify as comprehensively as possible the adopted method of intermediate foundation on CMC columns, the designer decided to execute six trial columns, two for each section of the building (see Fig. 8), namely at axes B8/21 and B8/16, columns P1 and P2, as well as at axes D4/20 and D2/14.

According to the test load design, all anchoring columns (four per tested column) were to be reinforced with IPE 120 steel sections. Identical reinforcement was adopted for the monitored columns. Initially, the monitoring included just four columns (designated as P3 to P4). After negotiations with the column contractor, approval was obtained for additional measurements on two tested columns (P1 and P2), which were not originally planned for systematic reinforcement and were reinforced using pipes available on-site.

Detailed monitoring covered the central section of the building, specifically the structure located at axes B12 - C7 / 10 - 22. Two columns, P1 and P2, were manually observed starting on October 12, 2015 then automatic monitoring was used for all columns to define the interaction of the CPRF foundation elements of central part of the building.

Comments 8:

Line 362 – A description of the columns load test would be helpful: for instance, how the load was applied, loading values, test procedure, identified measured quantities, and the test setup. It becomes clear later that topographic measurements were used. What instruments were used, and what was their sensitivity?

Response 8:

TEST PROCEDURE

The tests were conducted from November 4 to November 17, 2015.
The test loading was performed in accordance with PN-83/B-02482 Bearing capacity of piles and pile foundations.
To conduct the tests for 6 columns, a structure composed of a main beam (2x IPE550) with a length of 8 m and two secondary beams (2x HEB300) with a length of 4 m was used. The connection between the secondary beams and the anchoring piles was made by welding, as specified in the design documentation (PW).

The load application was performed using a setup consisting of: (a) 1 hydraulic cylinder (serial number 1) with a nominal force of 2000 kN, and an electric-powered hydraulic pump with a pressure gauge. (b) The column displacement was monitored using three dial gauges with a range of 50 mm and a readout accuracy of 0.01 mm. (c) For controlling the displacement of the anchoring columns, a precision leveling method was applied.

The test loads were performed in two stages:

Stage 1:
The load was incrementally increased until reaching a force approximately equal to the calculated design load (Q).
After reaching the force close to the calculated design load (Q), the column was unloaded to zero, and the permanent deformation was stabilized for 10 minutes.

Stage 2:
Following the Reinforcement Designer’s suggestion, the column was loaded to a force equal to 200% of the calculated design load. The column was then unloaded, and after 10 minutes, the permanent deformation was recorded.

Table: Load Test Results for P1

Stage

Force (kN)

Pressure (MPa)

Time (min)

Readings (mm)

Average (mm)

Increment (ds) (mm)

Settlement (mm)

Table: Load Test Results for P2

Stage

Force (kN)

Pressure (MPa)

Time (min)

Readings (mm)

Average (mm)

Increment (ds) (mm)

Settlement (mm)

Comments 9:

Line 402 – In addition to the hypothesis raised by the authors regarding local cross-section reduction (leading to increased deformation), the possibility of eccentricity in the metal tube inside the cross-section should not be ruled out, as this could result in measuring possible pile bending.

 Response 9:

The explanation for the higher force observed near column P2/2, i.e., the increased stress in the column cross-section, may indeed result from the eccentric positioning of the pipe, leading to bending of the column. Additionally, increased loading of the column due to negative skin friction cannot be excluded.

Comments 10: (done)

The lack of recovery of the surface settlement of column P1 at the end of the load test (Figure 20) could be better explained. Part of this settlement (around half) can be attributed to the axial deformation observed in each of the piles (estimated at around 5 mm). At the end of unloading, the deformation recovery in both piles P1 and P2 is not very different (Fig. 17a and Fig. 18a). At least this amount of settlement should have been recovered similarly in both piles. If, as stated (line 432), “the side surface counteracts the column (P1) from rising,” then the axial deformation of the column should remain, which is not the case.

Moreover, regarding the results in Figure 20, no explanation is given as to why, during the mid-test unloading, the settlement of P1 is recovered similarly to P2, whereas the same does not occur at the end of the test. Was there a reference error in measuring the settlement of P1 at the end?

 Response 10:

The reviewer is correct in stating that, when comparing the deformation results recorded in columns P1 and P2 (see Figures 17a and 18a), they are of the same order, i.e., approximately 50 microstrains. A different behavior was observed in Figure 20 – after removing the load, there was no recovery of vertical deformation measured at the head of column P1. To clarify the matter, the results of all six load tests were compared. For columns numbered 50 (in axis D4/20) , 126 (in axis D2/14), 329 (P2), 255 (P1), 434 (in axis D8/21), and 411 (in axis B8/16), the recovery of vertical deformation (the difference in settlement between the last load and unload step) is as follows: -4.60 mm, -3.82 mm, -6.71 mm, -0.46 mm, -5.06 mm, -3.99 mm, respectively.

Clearly, the measurement in column P1 differs from the other five, which indicates a high probability of an error in reading the vertical deformation.

Comments 11:

The sensor used (Geokon 4100) compensates for thermal effects when applied to steel. According to the manufacturer's technical specifications:

"If attached to a steel structure, the thermal coefficient of expansion of the steel vibrating wire inside the instrument is the same as that for the structure. This means that no temperature correction for the measured strain is required when calculating load-induced strains."

Response 11:

The method of estimating the forces in the column was described in line 286, chapter 2.4.2. “Assessment of compressive forces in CMC columns”.

Line 292 : “It was assumed that there is full adhesion between the steel profile and the concrete, ensuring compatibility of deformations. The transformed cross-sectional areas Acs were determined for columns with an IPE120 profile and a tubular profile, taking into account the ratio of the modulus of elasticity of steel Est = 200 GPa and concrete Ecm = 21.7 GPa.”

Comments 12:

This raises a question because the deformation is imposed by the concrete, which has a slightly lower linear thermal expansion coefficient. This aspect is omitted in lines 473 and following. The strain correction likely takes into account this behavior of the sensor used. I suggest clarifying how the strains that generate stresses are derived in Figure 20.

 Response 12:

As written in line 417 :” …the load-settlement relationship determined for their heads…” (Figure 20).

The graph in Figure 20 was prepared based on the results of the test loading, i.e., with knowledge of the force applied to the column head and the settlement measured by three dial gauges with a range of 50 mm and a readout accuracy of 0.01 mm (see comments 8 and Load Test Results for P1 mentioned above).

Comments 13:

Line 536 – The foundation slab description in Chapter 2 omits its plan dimensions. It is presumed that the 3.5 m × 3.5 m comes from somewhere, but it’s unclear where the 7.75 × 10.30 m dimension comes from.

Response 13:

The area defined by dimensions 7.75 m × 10.30 m has not been precisely described. It should be clarified that the geometry of the area scrutinized in chapter 3.4 Distribution of force on the elements of the combined piled raft foundation (line 492) has been defined, taking into account half the span between the center of the depression with dimensions of 3.5 m x 3.5 m and the subsequent neighboring structural elements.

Line 522 and 523 :

The area of the slab within the column, where the recess of the foundation slab with an area of 3.5 m × 3.5 m is located,…

Comments 14:

Line 573 – In Figure 24, how do the authors justify the continuous increase in pile forces 600 days after construction? These forces were calculated from strain data. I estimate that the stresses imposed on the pile concrete approach 10 MPa. At this stress level, shouldn't a creep model for concrete be included? Couldn’t this apparent increase in forces be partly explained by concrete creep?

Still on this subject:

Response 14:

As indicated in Table 1, the building's shell structure was completed approximately 200 days after the activation of the automatic measurement system. The shell structure (according to information provided by the structural designer) constitutes 50% of the total load. This means that in the subsequent days of measurement, the building's load was gradually increased. Measurements were concluded after approximately 800 days, assuming that the applied load reached 80% of the total load (excluding variable loads).

The reviewer's suggestion is indeed valid. The article presents a straightforward interpretation of the results obtained. The forces recorded in the columns, based on the conversion of measured deformations, were compared with the results from the FEM analysis (ZSoil). In the FEM model, only a simple elastic model for concrete was used. A more in-depth analysis could be conducted in line with the reviewer's comment, i.e., using a constitutive model for concrete that accounts for creep.

Comments 15:

Line 747 – This comment regarding continued construction work and increasing column loads should appear earlier, when Figure 24 is presented.

Response 15:

Thank you for your suggestion. I reviewed the text.

Comments 16:

Note that this increase in pile load is not accompanied by a similar increase in the load cell readings (see Figure 23), which again raises the question of whether the increase in forces in Figure 24 is real.

Response 16: 

To address the reviewer's question/concerns, a comparative analysis was conducted on the load distribution among the CPRF foundation elements, based on readings taken on days 251 and 788 of monitoring. On day 251, the forces transferred through the subsoil in the deepened area (based on S8), the area under the slab outside the deepening (based on S7), and through the 7 columns (based on P1/3) were 444 kN, 1190 kN, and 3860 kN, respectively.

On day 788, the corresponding forces were 612 kN, 1525 kN, and 6083 kN.

Thus, the total load transferred through the subsoil was 1635 kN after 251 days and 2138 kN after 788 days of monitoring, representing 29.7% and 26% of the total load, respectively. Comparing the load increase from day 251 to day 788, the load carried by the subsoil increased by 23.6%, while the load in the columns increased by 36.5%. The greater increase in column load can be attributed to the higher stiffness of the column supports compared to the subsoil.

Comments 17:

Line 597 – I suggest a better framing of the comparison being made between the “plate-pile system” and the “single-layer system.” I got a bit lost in the comparison.

Response 17:

It should be corrected :

The course of the axial force along the column for the last reading leads to the conclusion that the side surface of the column in the plate-pile system works weaker than in the single-layer system.

The course of the axial force along the column for the last reading leads to the conclusion that the shaft of the column in the CPRF system works weaker than in the single working column.

Reviewer 4 Report

Comments and Suggestions for Authors

This paper provides valuable insight into the behavior of Combined Piled Raft Foundations (CPRF), particularly under long-term monitoring for deformation and stress distribution. The authors effectively utilize finite element method (FEM) modeling to complement field data, both for validating sensor readings and extrapolating missing data. While the manuscript is generally well-written, several grammatical and structural revisions are recommended to enhance clarity. The literature review addresses several key studies in detail, each presented in an individual paragraph. However, its overall length can be reduced by summarizing overlapping information and consolidating redundant content. Following are some additional comments.

  • Page 8, Line 211: Correct the typo in the temperature range.
  • Page 12, Line 305: The forces were calculated under the assumption of constant column stiffness. This assumption may not hold over time due to creep, shrinkage, or material degradation of concrete. Although the impact appears minor in the current analysis, a brief paragraph justifying this assumption should be included. The authors are also encouraged to perform a sensitivity analysis to assess the influence of variable stiffness on the results, which would strengthen the manuscript.
  • Page 15, Figure 17 and Figure18: Include the time duration in hours on the x-axis for better interpretability of the results.
  • Page 21, Line 555-566: The paper notes that several sensors failed and stopped recording data after a certain period. Please elaborate on any known reasons for these sensor malfunctions. Furthermore, clarify the level of uncertainty associated with FEM-based interpolation of the missing data, including any error estimates or assumptions made.
  • Page 32, Line 857-859: Please provide an explanation for the existence of dead zone in middle columns and why it is only linked with the small changes in the axial force.

Author Response

This paper provides valuable insight into the behavior of Combined Piled Raft Foundations (CPRF), particularly under long-term monitoring for deformation and stress distribution. The authors effectively utilize finite element method (FEM) modeling to complement field data, both for validating sensor readings and extrapolating missing data. While the manuscript is generally well-written, several grammatical and structural revisions are recommended to enhance clarity. The literature review addresses several key studies in detail, each presented in an individual paragraph. However, its overall length can be reduced by summarizing overlapping information and consolidating redundant content. The following are some additional comments.

Comments 1:

Page 8, Line 211: Correct the typo in the temperature range.

Response 1:

I corrected it.

Comments 2:

Page 12, Line 305: The forces were calculated under the assumption of constant column stiffness. This assumption may not hold over time due to creep, shrinkage, or material degradation of concrete. Although the impact appears minor in the current analysis, a brief paragraph justifying this assumption should be included. The authors are also encouraged to perform a sensitivity analysis to assess the influence of variable stiffness on the results, which would strengthen the manuscript.

Response 2:

Thank you for your valuable comment. We acknowledge that the assumption of constant column stiffness over time may not fully capture long-term effects such as creep, shrinkage, or material degradation in concrete. The context of our study was to present a straightforward interpretation of the results obtained under the assumption of constant column stiffness. The forces recorded in the columns, based on the conversion of measured deformations, were compared with the results from the FEM analysis (ZSoil). In the FEM model also only a simple elastic model for concrete was used.

The authors are aware of certain limitations in accurately estimating the time-dependent behavior and stiffness changes of concrete columns. Nonetheless, a deliberate decision was made to perform an initial, simplified estimation of forces in the columns, with the primary aim of illustrating the load-sharing mechanism between the components of the CPRF system.

We agree with the reviewer that, for a more precise assessment of column behavior, it would be necessary to account for the long-term degradation of concrete stiffness due to effects such as creep and shrinkage. Performing such an analysis and comparing the results with the simplified, time-independent model would help evaluate the sensitivity of column force predictions to these time-dependent effects.

It is important to emphasize that a comprehensive consideration of all time-dependent effects in concrete can be very challenging. While individual phenomena—such as creep—can be analyzed separately using the common approach which is to use a creep coefficient, taken from standards such as Eurocode 2 or from empirical models in the literature (), simultaneously accounting for creep, shrinkage, and material degradation is complex and may exceed the scope of a simplified analysis. Each of these effects influences the stiffness and stress distribution differently and may require advanced rheological or numerical modeling for accurate assessment.

The authors believe that performing a sensitivity analysis is feasible; however, given the large amount of data obtained, it would be quite labor-intensive and time-consuming. It could serve as the basis for a separate publication.

Comments 3:

Page 15, Figure 17 and Figure 18: Include the time duration in hours on the x-axis for better interpretability of the results.

Response 3:

x-axis: Time duration in hours (approximately 4 hours)

              a)

              b)

Figure 17. a) Deformation of column P1 and b) axial force in column P1 during load test

a)

              b)

Figure 18. a) Deformation of column P2 and b) axial force in column P2 during load test

Comments 4:

Page 21, Line 555-566: The paper notes that several sensors failed and stopped recording data after a certain period. Please elaborate on any known reasons for these sensor malfunctions. Furthermore, clarify the level of uncertainty associated with FEM-based interpolation of the missing data, including any error estimates or assumptions made.

Response 4:

The lack of results observable between days 520 and 560 of monitoring is due to a break in data readings caused by the service provider, which resulted from the expiration of the initial monitoring service agreement. Once a new contract was signed, the readings were resumed.

A different issue is the absence of results for sensors numbered P2/3 and P6/1, which were damaged, for example, during installation. Yet another issue concerns the lack of readings in sensors numbered P4/3 and P3/2, where it should be noted that due to excessive stretching or shortening of the string in the sensor—i.e., going beyond the measurement range—the sensors did not provide valid data within the expected range of strain/force. Such results were deliberately removed from the force vs. time chart. Once the readings returned to the valid measurement range, the data was again included in the chart.

The authors made every effort to accurately model the layered soil and the building structure with a piled-raft foundation.

The forces recorded in the columns, based on the conversion of measured deformations, were compared with the results from the FEM analysis (ZSoil). In the FEM model, only a simple elastic model for concrete was used. A more in-depth analysis could be conducted in line using a constitutive model for concrete that accounts for creep.

The level of uncertainty can be assessed based on Figure 29, which compares the results of a test loading for column P2 with the simulation from the FEM analysis in ZSoil. Despite considerable efforts to model the layered soil medium accurately, a very good agreement between the presented results could not be achieved.

Comments 5:

Page 32, Line 857-859: Please provide an explanation for the existence of dead zone in middle columns and why it is only linked with the small changes in the axial force.

Response 5:

The authors noted that a small change in axial force along the length of the column indicates the presence of the so-called dead zone.

Directly beneath the slab in CPRF foundations, a reduction in shaft friction mobilization along the column is observed — the formation of a so-called 'dead zone' — to a depth that depends on the spacing and mutual arrangement of the columns, as well as the magnitude of foundation settlement.

Line 606 – should be

The observations led to the conclusion that the statement about the existence of the so-called "dead zone" that typically occurs at small axle spacing between columns of 3D should be considered correct.

Explanation of so-called dead zone was explained in lines 608 – 616.

In their work Hanisch, Katzenbach, and König (Kombinierte Pfahl-Plattengründungen, Ernst and Sohn, Berlin, 2002 indicated that the shaft friction distributions they presented for small center-to-center spacings, i.e., 3D, suggest that for a small range of settlements (0.03D), the raft in a piled-raft foundation negatively influences the distribution of unit shaft friction along the pile in the zone directly beneath the raft (up to 6 m, or 1/5L for a central pile, and up to 25 m, or 0.83L for corner and edge piles). For greater settlements (0.10D), the behavior of the piles in a group and in the piled-raft foundation is very similar, although the performance of the central pile in the piled-raft foundation is noticeably better. The reduction in shaft friction mobilization directly beneath the raft results from the lack of relative displacement between the pile shaft and the surrounding soil.

Reviewer 5 Report

Comments and Suggestions for Authors

The comments and suggestions are provided in the attached document.

Author Response

The paper presents a detailed investigation into the load transfer mechanism in a Combined Piled Raft Foundation (CPRF) system using instrumented Controlled Modulus Columns (CMC). The methodology combines field measurements, numerical simulations (ZSoil), and monitoring of stress and strain within the CMC columns and under the foundation slab. The study contributes valuable insights into the actual behavior of CMC columns embedded in a CPRF context and illustrates the potential mismatch between isolated load tests and in-situ performance. However, several issues related to clarity, consistency, and technical completeness should be addressed to improve the quality and impact of the paper:

Comments 1:

  1. The parameters used in Equation (1) are not clearly defined immediately before or after the equation. To improve readability and self-containment, all variables should be explicitly explained at the point of introduction.

Response 1:

Ncol=Normal force of column, Acs= Cross sectional area of column, Ecm=elastic modulus of sandstone aggregate and =the change in deformation measured by the sensor

Comments 2:

  1. In the caption of Figure 13, both steel pipes and IPE120 profiles are mentioned as structural elements used to transfer load into the columns. However, the paper does not explain why two different profile types were used or under what conditions. This raises questions about consistency in instrumentation or load application.

Response 2:

In order to verify as comprehensively as possible the adopted method of intermediate foundation on CMC columns, the designer decided to execute six trial columns, two for each section of the building (see Fig. 5), namely at axes B8/21 and B8/16, columns P1 and P2, as well as at axes D4/20 and D2/14.

According to the test load design, all anchoring columns (four per tested column) were to be reinforced with IPE 120 steel sections. Identical reinforcement was adopted for the monitored columns. Initially, the monitoring included just four columns (designated as P3 to P4). After negotiations with the column contractor, approval was obtained for additional measurements on two tested columns (P1 and P2), which were not originally planned for systematic reinforcement and were reinforced using pipes available on-site.

Line 294-297

“It was assumed that there is full adhesion between the steel profile and the concrete, ensuring compatibility of deformations. The transformed cross-sectional areas Acs were determined for columns with an IPE120 profile and a tubular profile, taking into account the ratio of the modulus of elasticity of steel Est = 200 GPa and concrete Ecm = 21.7 GPa.”

Figure 13 shows that it is during the construction phase.

Comments 3:

  1. In line 500, the authors refer to Figure 2 to illustrate the arrangement of sensors S7, S8, and S9 and the slab recess. However, Figure 2 does not depict these details.

Response 3:

Figure 23 should be compared with Figure 5

Comments 4:

  1. In Section 3.1, the phrase "in the graphs below" is used, yet Figure 15 appears above the text. It would be clearer to refer to "Figures 15 and 16" or use "the following graphs" to avoid confusion about figure positioning.

Response 4:

The dotted lines in the Figure 15 and 16 were introduced only to increase the Figures' readability because the intermediate periods' strain functions are unknown.

Comments 5:

  1. In lines 543 and 789, the paper refers to "Figure 170" and "Figure 177," neither of which exist in the manuscript. This numbering should be corrected.

Response 5:

Figure 170 =figure 25

Figure 177= figure 29

Comments 6:

  1. The acronym "FPP column" is first introduced in line 601 but is not defined anywhere in the paper. The paper previously uses the term CPRF (Combined Piled Raft Foundation) in the introductory and theoretical sections. It is unclear whether FPP is intended as a synonym for CPRF or refers to a specific variation. This should be explicitly clarified to maintain terminological consistency.

Response 6:

 FPP means CPRF, It is replaced in the text.

Comments 7:

  1. In lines 768–769, the text mentions axial force values estimated at a depth of 14.48 m based on FEM simulation (ZSoil). However, it is not explicitly stated that this depth corresponds to the point labeled “P2/0” in the legend of Figure 31. Since this value supplements the measured data and is directly plotted in the graph alongside experimental points, it would improve clarity if the correspondence between “P2/0” and the 14.48 m FEM result were explicitly mentioned in the text.

Response 7:

Line 768 – should be corrected : Additionally, the values of axial force at a depth of 14.48 m estimated in ZSoil were plotted with label ZSoil P2/0.

Comments 8:

  1. In Figures 23 and 24, a noticeable data gap occurs between approximately days 520 and 560. While sensor malfunctions are briefly mentioned in Section 3.5, the paper does not explicitly acknowledge this widespread interruption in the graphs. It is recommended that this be highlighted in the figure legends or discussion, as the discontinuity may confuse readers or appear as missing information.

Response 8:

The lack of results observable between days 520 and 560 of monitoring is due to a break in data readings caused by the service provider, which resulted from the expiration of the initial monitoring service agreement. Once a new contract was signed, the readings were resumed.

A different issue is the absence of results for sensors numbered P2/3 and P6/1, which were damaged, for example, during installation. Yet another issue concerns the lack of readings in sensors numbered P4/3 and P3/2, where it should be noted that due to excessive stretching or shortening of the string in the sensor—i.e., going beyond the measurement range—the sensors did not provide valid data within the expected range of strain/force. Such results were deliberately removed from the force vs. time chart. Once the readings returned to the valid measurement range, the data was again included in the chart.

Comments 9:

  1. In line 608, the sentence "Columns of the 3D row This situation can be explained by..." lacks proper punctuation and reads as two disconnected fragments. This grammatical inconsistency should be corrected for clarity.

Response 9:

Thank you for the recommendation. I it corrected and highlighted in the text.

Comments 10:

  1. The column load test is mentioned in Section 3.1 without any accompanying explanation or reference. Since the procedure is described in detail in Section 3.2, the paper should explicitly guide the reader to this later section (e.g., "see Section 3.2") for clarity and continuity.

Response 10:

It is corrected in the text.

Comments 11:  

  1. Although shaft shortening is attributed exclusively to Figure 18 in line 349, a similar phenomenon is clearly visible in Figure 17. The authors should consider updating the reference to include both Figures 17 and 18, as the behavior appears in both columns.

Response 11:

It is corrected according to the suggestions and comments.

Comments 12:

  1. In Section 3.2, the paper describes unloading of the columns after the fifth loading stage, which is visible in Figures 17 and 18. However, the rationale for this decision is not explained. Including the reason for unloading (e.g., evaluating elastic recovery, calibrating measurement devices) would enhance the reader's understanding of the testing procedure.

Response 12:

After the fifth stage the reason of the unloading is to evaluate elastic recovery, calibrating measuring devices.

Regarding the results in Figure 20, during the mid-test unloading, the settlement of P1 is recovered similarly to P2, whereas the same does not occur at the end of the test.

When comparing the final deformation results recorded in columns P1 and P2 (see Figures 17a and 18a), they are of the same order, i.e., approximately 50 microstrains. A different behavior was observed in Figure 20 – after removing the load, there was no recovery of vertical deformation measured at the head of column P1. To clarify the matter, the results of all six load tests were compared. For columns numbered 50 (in axis D4/20) , 126 (in axis D2/14), 329 (P2), 255 (P1), 434 (in axis D8/21), and 411 (in axis B8/16), the recovery of vertical deformation (the difference in settlement between the last load and unload step) is as follows: -4.60 mm, -3.82 mm, -6.71 mm, -0.46 mm, -5.06 mm, -3.99 mm, respectively.

Clearly, the measurement in column P1 differs from the other five, which indicates a high probability of an error in reading the vertical deformation.

Comments 13:

  1. In line 818, the depth of 14.5 m is mentioned as the location of additional FEM data, while in line 769 the value is given as 14.48 m. Although the difference is negligible, it is recommended to standardize this value throughout the text to avoid potential confusion.

Response 13:

14.48m is recommended.

Comments 14:

  1. I would kindly recommend that the introductory part of the paper include a description of the monitored building, accompanied by an illustration of the structure and representative cross-sections. Presenting the geometry, number of floors, and the relationship between the raft slab and the columns at the outset would give readers a clearer understanding of the structural system and significantly improve the overall clarity of the study.

Response 14:

Thank you for your recommendation and suggestions. It was modified and included within the manuscript.

Overall, the paper presents valuable field data and modeling results, but several inconsistencies and unexplained choices limit the clarity and completeness of the current version. A clearer structure, consistent terminology, and more thorough figure annotations would significantly enhance the paper's accessibility and academic impact.

Thank you for your contribution to the field of geotechnical monitoring and foundation engineering.

Round 2

Reviewer 1 Report

Comments and Suggestions for Authors

This paper has responded well to previous comments and is ready for publication.

Reviewer 3 Report

Comments and Suggestions for Authors

The responses provided by the authors and the changes made to the text generally address the questions raised and clearly complement the original content.

I congratulate the authors on the work they have developed.

Reviewer 5 Report

Comments and Suggestions for Authors

I appreciate your dedication in carefully considering and implementing the feedback. The revisions have made a noticeable difference, and I have no further comments. Thank you for your excellent work.